# Urbanization and agricultural intensification destabilize animal communities differently than diversity loss

Théophile Olivier [1✉], Elisa Thébault[2], Marianne Elias[3], Benoit Fontaine[1] & Colin Fontaine[1]

Despite growing concern over consequences of global changes, we still know little about potential interactive effects of anthropogenic perturbations and diversity loss on the stability of local communities, especially for taxa other than plants. Here we analyse the relationships among landscape composition, biodiversity and community stability looking at time series of three types of communities, i.e., bats, birds and butterflies, monitored over the years by citizen science programs in France. We show that urban and intensive agricultural landscapes as well as diversity loss destabilize these communities but in different ways: while diversity loss translates into greater population synchrony, urban and intensive agricultural landscapes mainly decrease mean population stability. In addition to highlight the stabilizing effects of diversity on ecologically important but overlooked taxa, our results further reveal new pathways linking anthropogenic activities to diversity and stability.

[1] Centre d'Ecologie et des Sciences de la Conservation, UMR 7204 MNHN-CNRS-Sorbonne Université, Muséum national d'Histoire naturelle de Paris, 43 rue Buffon, 75005 Paris, France. [2] Sorbonne Université, CNRS, IRD, INRAE, Université Paris Est Créteil, Institute of Ecology and Environmental Sciences of Paris (iEES-Paris), Paris, France. [3] Institut de Systématique, Evolution, Biodiversité, MNHN-CNRS-Sorbonne Université-EPHE-Université des Antilles, Muséum national d'Histoire naturelle de Paris, 43 Rue Buffon, 75005 Paris, France. ✉email: theophile.olivier@gmail.com

Temporal stability, that is to say the level of variation through time, of biomass or abundance can vary greatly across local communities, and the causes of such variations remain poorly understood[1]. The stability of plant and animal community abundances is important for the maintenance of ecosystem processes and services over time, as these communities are involved in key functions such as primary and secondary productions, pollination, and pest control[2]. In the past decades, research on the stability of community properties and ecosystem processes has mainly focused on consequences of ongoing biodiversity loss, often on plant communities[2,3], revealing a negative effect of diversity loss on the temporal stability of communities[4–6]. This destabilizing effect of diversity loss appears mainly related to lower asynchrony among population dynamics in species poor communities[5,6]. Recently, a few studies brought to light the importance of other major anthropogenic changes, such as nutrient eutrophication and climate warming, on plant community stability and associated primary production[7–10]. While some of these studies highlight that anthropogenic changes affect ecosystem stability mainly via changes in biodiversity[7,11], other studies suggest independent effects of diversity and environmental changes on stability[12]. Resolving this discrepancy is key to our understanding of the mechanisms by which global changes affect the stability of ecosystem functions and services[13], and therefore to our ability to mitigate adverse effects. Furthermore, to our knowledge existing studies mainly focused on plant communities, resulting in a knowledge gap regarding animal communities.

At a global scale, the conversion and degradation of habitats related to human activities are recognized as major drivers of local diversity loss, urban and intensive agricultural land-use being most detrimental[14]. Despite the paramount importance of habitat degradation on biodiversity, only a few recent studies have investigated the consequences of land-use intensity on the temporal stability of population abundances or community total abundances[15,16], i.e., the total number of individuals present in a community, across all species of this community.

To investigate the mechanisms by which diversity loss and habitat degradation affect community stability, we analyze the inter-annual abundance fluctuations of 152 bat, 269 bird and 130 butterfly communities across France, monitored following standardized protocols over six, 17 and 11 years, respectively (Fig. 1, Supplementary Table 1, see "Methods"). The three taxonomic groups are not monitored on the same sites as data comes from three independent citizen science programs (see "Methods"). While bats and insectivorous birds are recognized as important for pest control[17,18], butterflies contribute to pollination[19], and frugivorous birds are essential for plant dispersal[18]. As such, understanding what determine the stability of these communities might be relevant to understand the stability of the functions and services they provide.

We analyze the landscape surrounding each sampling site looking at its composition, heterogeneity, and the level of agricultural inputs used (see "Methods"). Using a principal component analysis on these data, we distinguish two independent habitat degradation gradients (Fig. 2). First an urban gradient opposing sites surrounded by urban and sealed soil areas to sites surrounded by semi-natural and agricultural landscape. Second, an agricultural intensity gradient, opposing sites within landscapes dominated by cropland areas with high agricultural inputs to sites surrounded by heterogenous landscape including higher proportion of woodland areas and seminatural open areas.

For each sites, we compute local Shannon index as species diversity and mean pairwise distance (MPD) as phylogenetic diversity (Supplementary Table 2, see "Methods"), as there is still debate on which aspect of diversity better relates to community

stability[20,21]. Because species in natural communities are not evenly abundant and then do not have the same contribution to the community stability, we weight the two diversity measures by species relative abundance. We then partition community stability, measured as the inverse of the coefficient of variation of community abundance across time, into a population asynchrony and a weighted mean population stability component[22] (Supplementary Table 2, see "Methods"). Those complementary components of community stability have different implications. Lower population stability increases species extinction risks[23], while lower population asynchrony reduces the insurance effect of diversity on the provision of ecosystem functions and services[24]. These two components of community stability involve partly different mechanisms. Population asynchrony strongly relates to the diversity of species responses to environmental variations[25] but also to species interactions such as competition[26]. Population stability is known to depend on environment variability[27] and on interactions among species such as competition or predation[28,29]. While both components appear to be key to the stability of experimental plant communities[4,5], their relative contributions to variations in stability of natural communities experiencing various perturbation regimes is poorly known. We use structural equation modeling (SEM) to quantify how both components of community stability are affected by local diversity and landscape composition, as well as to disentangle the direct and indirect (i.e., mediated by diversity changes) effects of landscape composition on community stability (see Methods). Our analysis reveals that habitat degradation and diversity loss destabilize ecological communities but in different ways: while diversity loss destabilizes communities by increasing population synchrony, the destabilizing effect of urban and intensive agricultural landscapes mainly comes from a decrease in population stability.

## Results and discussion

**Diversity loss and habitat degradation effects on stability.** Our results show that communities with low diversity or located in anthropogenic landscapes exhibit lower temporal stability than more diverse communities or communities surrounded by more semi-natural and heterogeneous landscapes (Fig. 3), with overall effects of similar magnitude between the effects of diversity and landscape composition (Table 1).

**Diversity loss effects through population asynchrony.** The SEM suggests that the loss of both species diversity and phylogenetic diversity destabilize bat, bird, and butterfly communities, and that this is mainly channeled by a decrease in population asynchrony (Table 1, Fig. 4, Supplementary Figs. 1–3). This extends classical results found for experimental plant communities[4,5] to animal taxa with very different natural history and thereby suggests that the positive biodiversity-stability relationship found for primary production applies to other functions and services provided by animal communities. Our results also bring new support to the phylogenetic insurance hypothesis[20], with a positive effect of phylogenetic diversity on population asynchrony found for bat and butterfly communities. This suggests that for these taxa, related species tend to share the traits involved in their response to environmental variations and perturbations[30], and that we can use phylogeny as a proxy to assess the dynamical response of their populations to such environmental variations and perturbations.

**Diversity loss effects through population stability.** The effects of species diversity and phylogenetic diversity on the weighted mean population stability are weaker and more contrasted among the three taxa (Table 1, Fig. 4, Supplementary Figs. 1–3),

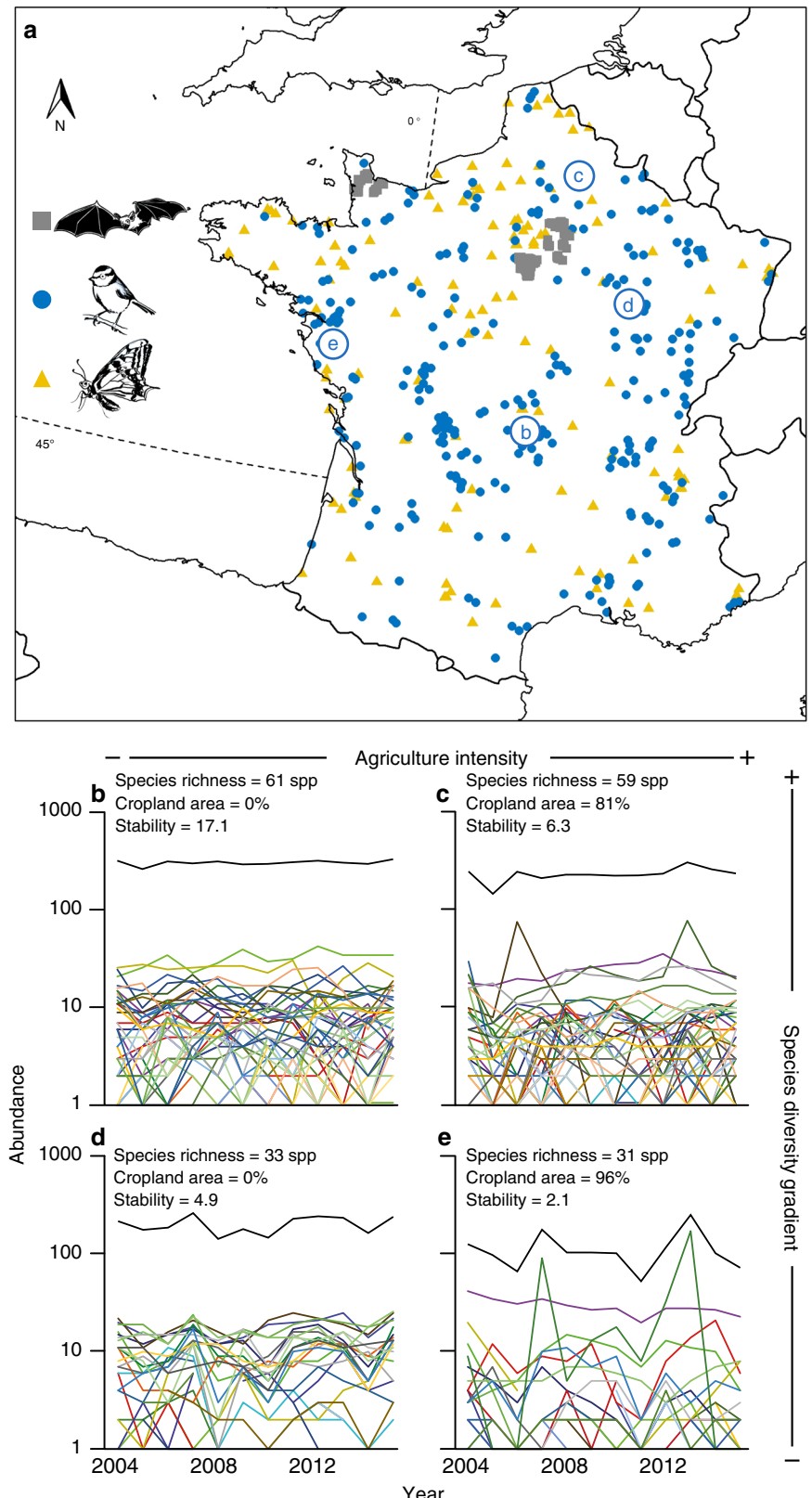

**Fig. 1 Datasets. a** Spatial distribution of the monitored bat, bird and butterfly communities across France. Circled letters correspond to the bird communities with contrasted levels of species richness and cropland area within buffer plotted in (**b–e**). Each colored line represents a given species abundance time series and black lines represent total community abundance time series. Species richness, cropland area within buffer, and stability (computed as the inverse of the coefficient of variation) of the total community abundance time series are given for each bird community.

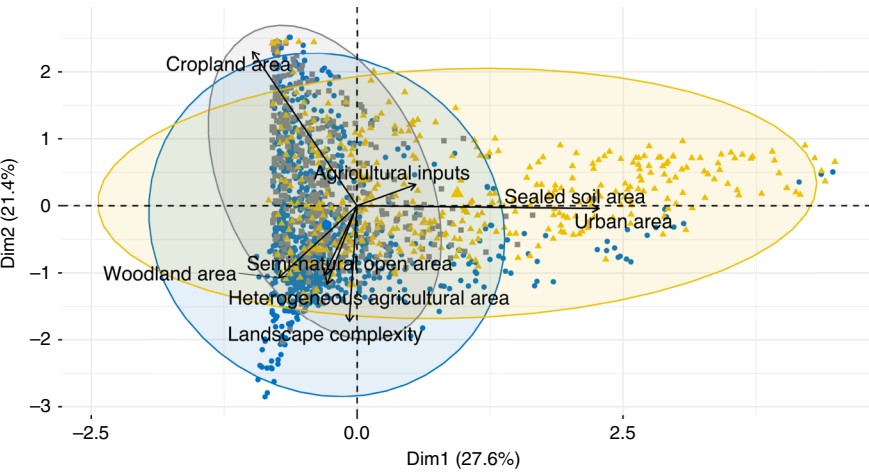

**Fig. 2 Principal component analysis performed on the landscape variables describing the study sites.** Each point corresponds to a site and a buffer size, colors correspond to the different taxa and each ellipse contains 95% of the sites of the corresponding taxa. Each axis is a linear combination of the variables describing the landscapes. Each arrow indicates the contribution to the two PCA axes of each variable describing the landscape. Gray squares correspond to sites with bat communities, blue circles to sites with bird communities, and yellow triangles to sites with butterfly communities.

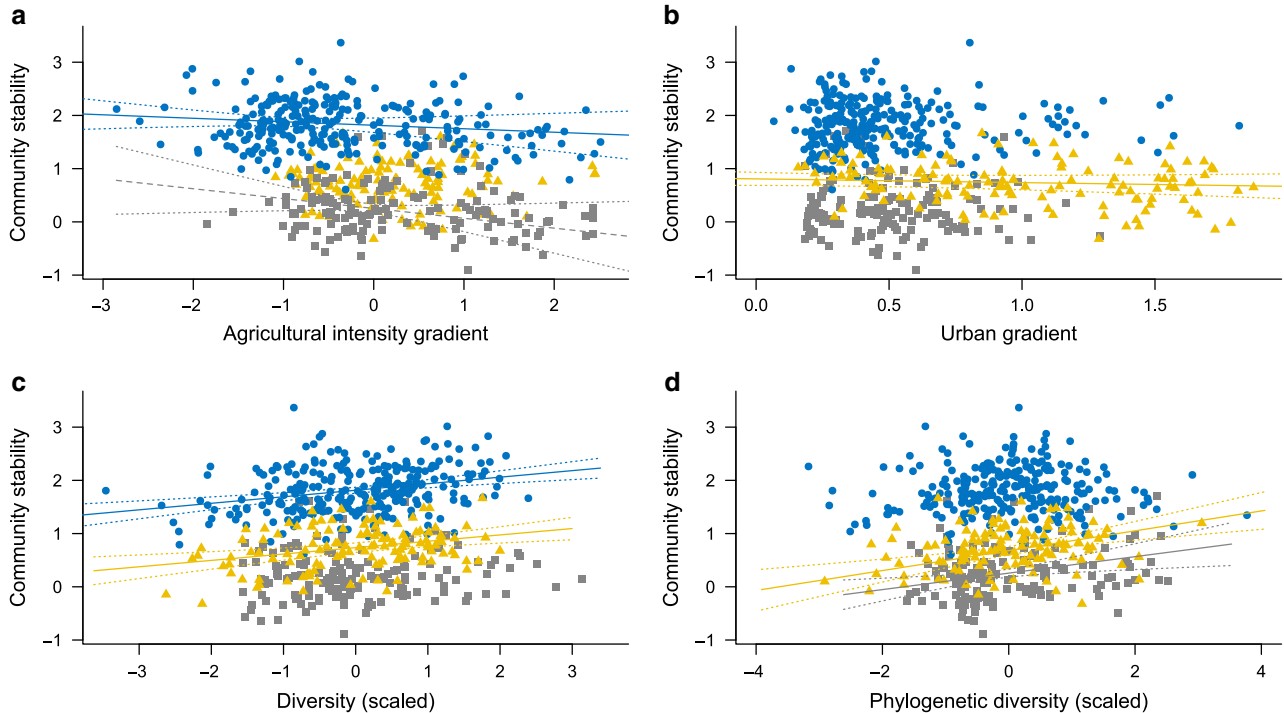

**Fig. 3 Regressions between community stability and landscape gradient or community diversity.** Panels refer to agricultural intensity gradient (**a**), urban gradient (**b**), species diversity (**c**), and phylogenetic diversity (**d**). Bats, birds, and butterflies are in gray squares, blues circles, and yellow triangles, respectively. Solid and dashed lines represent model predictions for significant and marginally significant ($p$ value = 0.06) relationships, respectively. Dotted lines represent the CI 95% of model predictions. For bats estimate = $-0.08$, $p$ value = 0.06 and estimate = 0.11, $p$ value = $4 \times 10^{-3}$ for (**a**) and (**d**), respectively. For birds estimate = $-0.10$, $p$ value = $3 \times 10^{-3}$ and estimate = 0.12, $p$ value = $8 \times 10^{-6}$ for (**a**) and (**c**), respectively. For butterflies estimate = $-0.17$, $p$ value = 0.01, estimate = 0.12, $p$ value = $5 \times 10^{-4}$, and estimate = 0.08, $p$ value = 0.02 for (**b**-**d**), respectively. Linear mixed-effects models are used for bats, and linear models for birds and butterflies. Diversity and phylogenetic diversity values are scaled so that all three taxa can be represented in the same plot.

reflecting previous theoretical and experimental findings on plants and plankton, which predict varying diversity–population stability relationships depending on the community context considered[28,29].

**Landscape composition effects through population stability.** Our results further highlight that two major drivers of habitat degradation, urbanization and agricultural intensification, decrease community stability mainly via a decrease in the weighted mean population stability, this for all three taxa (Table 1, Fig. 4, Supplementary Figs. 1–3). The mechanisms linking intensive agricultural landscapes and/or urban areas to lower weighted mean population stability may be related to the diversity and availability of resources for species[16,31]. Overall,

**Table 1 Strength of the different paths by which diversity and habitat degradation affect the stability of bat, bird, and butterfly local communities.**

| Effects on community stability | Bat | Bird | Butterfly |
|---|---|---|---|
| Diversity | | | |
| Total effects | 0.270 | 0.264 | 0.487 |
| Diversity effects | 0.116 | 0.264 | 0.180 |
| Phylogenetic diversity effects | 0.154 | NS | 0.307 |
| Effects via population stability | −0.132 | NS | 0.143 |
| Effects via population asynchrony | 0.402 | 0.264 | 0.344 |
| Habitat degradation | | | |
| Total effects | −0.20 | −0.291 | −0.134 |
| Urban effects | 0.004 | −0.043 | −0.247 |
| Agricultural intensity effects | −0.204 | −0.248 | 0.112 |
| Effects via diversity and phylogenetic diversity | −0.026 | −0.092 | 0.013 |
| Effects via population stability | −0.175 | −0.198 | −0.260 |
| Effects via population asynchrony | NS | NS | 0.112 |

The strength of each path is calculated from the standardized coefficients of the structural equation models, multiplying coefficients along a path and summing the results over the different paths. Total effects refer to the sum of the effects by which species richness and phylogenetic diversity loss, or urban gradient and agricultural intensity gradient, affect community stability. Effects via a variable are the sum of the effects of diversity loss/habitat degradation on community stability that are channeled by a direct effect on this variable. NS stands for nonsignificant effects.

degraded habitats with less diverse and abundant resources, such as food supply, nesting, or breeding site and hunting territory, are likely to result in smaller populations which, according to Taylor's law[32], might decrease their stability. Such a link between abundance and stability is supported for bats and butterflies as their mean population abundances decline faster than their variances in degraded habitats (Supplementary Fig. 4, Supplementary Fig. 5), resulting the observed declines in weighted mean population stability (Fig. 4). For birds, however, the destabilizing effect of agriculture is not related to decreased mean population abundances, but to an increase in the temporal variance of populations in areas with intensive agricultural land-use (Supplementary Fig. 4). Such an effect suggests another potential mechanism where intensive agricultural lands tend to increase the variability of available resources, such as crop pests, and/or tend to have more frequent and/or stronger pulse perturbations, such as pesticide application.

Our results also suggest differences among taxa in their susceptibilities to different types of habitat degradation. While butterfly community stability is mainly impacted by urban areas, bird and bat communities are mostly destabilized in intensive agricultural landscapes (Fig. 3a, b, Table 1). This is coherent with previous studies on butterflies[33] and birds[34,35], but could also stem from different levels of habitat degradation as butterfly communities were sampled in more urbanized areas than were bird and bat communities (Fig. 2).

**Direct and indirect landscape effects on community stability.** Although habitat degradation directly affects the diversity of all three taxa, the effects of habitat degradation on community stability via population stability are 6.7, 2.2, and 20 times stronger than those mediated via its effects on both species diversity and phylogenetic diversity for bats, birds and butterflies, respectively (Table 1, Fig. 4, Supplementary Figs. 1–3). These results contrast with previous findings on plants, butterflies, and birds indicating that anthropogenic perturbations effects on community stability were mainly channeled by direct changes in diversity[5] or population asynchrony[15]. Those contrasted findings across studies may stem from the fact that habitat degradation affects the diversity of bat, bird, and butterfly communities in slightly more complex and contrasted ways than it affects their stability. Landscapes dominated by intensive agriculture or urban areas

decrease the species diversity of bird community, consistently with previous findings[34]. While landscapes dominated by urban area sharply decrease butterfly species diversity as already found[36], they increase the species diversity of bat communities and the phylogenetic diversity of bird and butterfly communities. Such positive effects have already been shown for moderate levels of urbanization, where urban exploiters and exotic species can mix with urbanophobe species[37,38]. Such results also echo previous findings highlighting complex patterns of species diversity variations along urban or agricultural intensity gradients associated with non-random changes in community composition[37,39].

**Limits and conclusion.** Here, we measured community stability at a relatively short-time scale (up to 6, 17, and 11 years for, respectively bat, bird, and butterfly communities), reflecting the time scale used in most studies on the relationship between diversity and community stability[6]. However, population and community temporal variability are known to increase with the considered time scale[40–42] and as such, our estimates of temporal variability might underestimate the full variability of the studied communities. While this should not affect the effects of landscape composition we found, and indeed our results are robust when compared with analyses on two subsets of our datasets with different time series durations (see "Methods" and Supplementary Figs. 7–9), longer time series would be required to estimate the full variability of the studied communities. Another limitation of our study is that we assessed habitat degradation at the landscape scale and did not account for local conditions, such as management practices, that could also affect community variability. For example, butterfly data were collected in private gardens with different management strategies that are known to affect the attractiveness for butterflies[43]. Accounting for such management practices as well as other local scale characteristics such as habitat heterogeneity that is also known to affect population stability[16] would improve our understanding of the determinant of community stability.

In summary, our results extend to various animal communities the classical diversity–stability relationship found for plants[5,6] and further uncover a population-level destabilizing effect of habitat degradation. Moreover, by increasing the risks of extinction through the destabilization of populations, habitat degradation may also enhance the negative effect of diversity loss

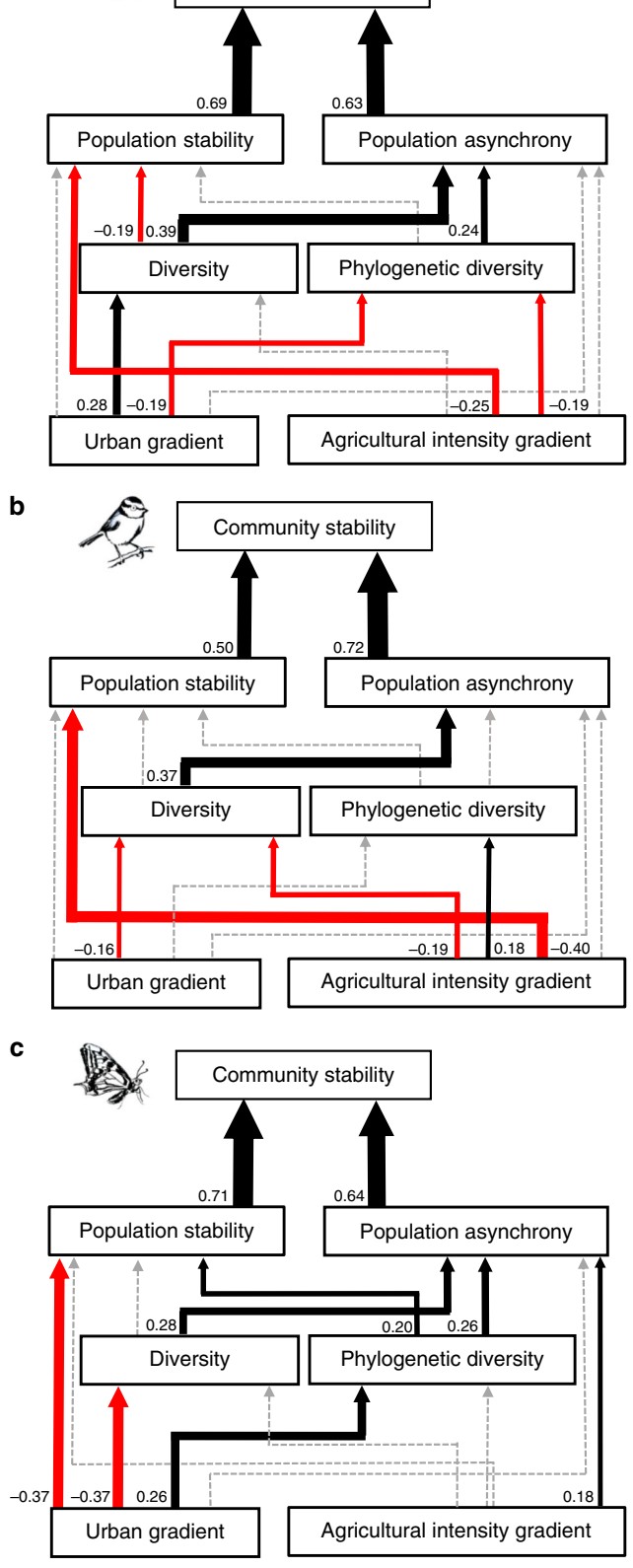

**Fig. 4 Effects of habitat degradation and local diversity on community stability estimated by structural equation models.** Panels refer to bat (**a**), bird (**b**), and butterfly (**c**) communities. Urban gradient and agricultural intensity gradient are independent variables while species diversity, phylogenetic diversity, population stability, population asynchrony, and community stability are dependent variables. Black and red arrows represent positive and negative relationships, respectively, with relationship strength depicted by arrow width. Solid and gray dashed arrows represent significant relationships and nonsignificant relationships, respectively (Supplementary Table 3, see "Methods"). Standardized coefficients are shown next to the arrow basis.

might act. Finally, by providing numerous long-term time series of local communities under real perturbation regimes, citizen science monitoring-programs emerge as a major tool to further our understanding of the dynamical consequences of current global change.

## Methods

**Community dynamics datasets**. Yearly butterfly, bird, and bat abundance data were obtained across France from nationwide citizen science monitoring schemes, part of the Vigie Nature program [http://vigienature.mnhn.fr/]. The monitoring sites are different among taxa as they depend on the residency of the volunteers and monitoring protocol used (see below).

For bat communities, we used data from the French bat-monitoring program [http://www.vigienature.fr/fr/chauves-souris], a citizen-science program running since 2006[44]. Briefly, volunteers record bat activity using ultrasound recorder while driving at a constant low-speed ($25 \pm 5$ km/h) along 30 km circuits. These circuits were chosen to be close to the volunteer residency, with low-traffic roads for security and representative of the different land-cover types in the area. These circuits are divided into ten 2-km transects where bats are recorded, separated by 1-km road portions where recording is not carried out. Surveys start 30 min after sunset and last approximately 1.5 h. Weather conditions have to be suitable for the survey to be carried out: no rain, wind speed below 7 m/s, temperature above 12 °C. Bat activity is recorded through echolocation calls with ultrasound detectors connected to a digital recorder. Volunteers were trained to classify echolocation calls to the most accurate taxonomic level using Syrinx 2.6[45]. Data validation was done by program coordinators at the Muséum national d'Histoire naturelle for recordings with uncertain identification. Following previous works (e.g.,[46]), the abundance of each bat species in a 2-km road transect was defined as the number of bat pass per species (a bat pass corresponds to a trigger of the bat detector in time expansion). We used data from transects surveyed between June 15th and July 31st (seasonal bat activity peak) from four to 6 years between 2006 and 2012. This represented a total of 152 local bat communities, where a total of 7 species were recorded (Supplementary Table 1).

For bird communities, we used data from the French Breeding Bird Survey [http://www.vigie-plume.fr/][47], a monitoring program relying on keen birdwatchers to count birds annually in a given plot. Plots are squares of $2 \times 2$ km² randomly selected by the national coordinator, within which the surveyor places 10 points separated by at least 300 m, in order to cover all the habitats present in the plot. Each plot is surveyed twice a year, the first session (to record nonmigrant birds and short distance migrants) between April 1st and May 8th, the second (to record trans-saharian migrants) between May 9th and June 30th, with at least 4 weeks between both sessions. Surveying dates must be the same (±5 days) every year, and counting takes place in the morning, starting 30 min after sunrise, with points always visited in the same order. At each point, the volunteer spends 5 min recording all birds seen or heard. Following previous work (e.g.)[47], yearly species abundance at a site was calculated as the sum of the maximum number of individuals detected per point over the two sessions. For this study, plots surveyed at least eight years between 2001 and 2017 were selected, representing 269 local bird communities and 75 common species for which the amount of data available allows an accurate estimation of population dynamic (Supplementary Table 1).

For butterflies, we used data from the French garden butterfly observatory [http://www.vigienature.fr/fr/operation-papillons][48]. Participants identify and count Lepidoptera in their own garden, from a closed list of 28 common species or species groups (27 butterflies and one common diurnal moth, *Macroglossum stellatarum*). Since some of the taxa targeted by this scheme group several look-alike species (species groups), we only kept the 13 butterflies and 1 moth identified at species level for our analyses (Supplementary Table 1). The temporal variability of the butterfly community restricted to these 14 species reflects that of the community including the abundance of the species groups (Supplementary Fig. 6). For each species, abundances are recorded monthly, as the maximum number of butterflies seen simultaneously during the month. Counting takes place from March to October, but for this study, gardens that had been monitored in July at least seven years between 2007 and 2018 were selected, except for the year 2014

on community stability. Besides advancing our understanding of the stability of animal communities and thereby the functions and services they deliver, our findings are also relevant for biodiversity conservation and management as they identify different pathways affecting community stability on which conservation policies

because of a crash of the database server, representing 130 local butterfly communities.

Our data selection presented and analyzed in the main text totalized 551 sites (Dataset 1), with variations in the number of years across sites and taxa. This leads to 65, 7, and 80 sites with time series of, respectively 4–6 years for bats; 3, 14, 37, 32, 29, 33, 49, 30, 30, and 12 sites with time series of, respectively, 8–17 years for birds; and 34, 12, 10, 17, and 57 sites with time series of, respectively, 7–11 years for butterflies.

To test the robustness of our results to the presence of gaps in the time series, we also ran our analyses on two subsets of the dataset that included only time series of the same duration and with no missing year. The first subset was restricted to the longest fully overlapping observation period with no gap common to all sites, leading to times series of 4, 8, and 7 years for bats, butterflies and birds, respectively (Dataset 2). The second one was restricted to sites having the longest fully overlapping observation period with no gap, leading to time series of 5, 12, and 11 years, for bats, birds, and butterflies, respectively. This last data selection procedure reduced the number of sites included to 87, 106, and 57, for bats, birds, and butterflies, respectively (Dataset 3). The analysis of the three datasets gave qualitatively similar results (Supplementary Figs. 7–9), confirming the robustness of our results to both the presence of gaps and the number of communities studied.

**Community species diversity and phylogenetic diversity.** For each local community, we estimated the species diversity as the exponential of the Shannon diversity index calculated from the summed yearly abundance across all years for each species seen in a site during the survey period. We assessed the phylogenetic diversity of each community using the MPD index weighted by species abundances[49]. As the weighted MPD was correlated to species diversity for the three taxa, all analyses were performed using the residuals of the weighted MPD against the species diversity (Supplementary Fig. 10).

The weighted MPD calculations were based on ultrametric molecular phylogenetic trees[50]. For bats we used the phylogenetic tree provided by Shi and Raboski[51] (Supplementary Fig. 11). For birds we extracted 1000 trees from the phylogeny published by Jetz et al.[52] and computed the Maximum Clade Credibility tree with branch lengths equal to the median of the branch lengths of the 1000 trees using TreeAnnotator 1.7.5[53] and without burnin. The resulting tree was well supported, with 93% of the nodes having a Bayesian Posterior Probability > 0.9 (Supplementary Fig. 12). For butterflies, we downloaded published sequences for the following genes: cytochrome oxidase c subunit 1 (COI, 657 bp), elongation factor 1 alpha (EF1α, 1239 bp), glyceraldehyde 3-phosphate dehydrogenase (GAPDH, 690 bp), ribosomal protein S5 (RPS5, 616 bp), wingless (wg, 402 bp) (Supplementary Table 4). Sequences were aligned using CodonCode Aligner 6.0.2 [http://www.codoncode.com], and the different genes were concatenated. Phylogenetic analyses were performed in a Bayesian framework using BEAST 1.8.1. The dataset was partitioned by gene. Unlinked GTR + Γ model of nucleotide substitution and uncorrelated lognormal clocks were implemented for all partitions. The MCMC analysis was run for 100 million generations, and sampled every 100,000 generation, which resulted in 1000 trees. After checking for convergence, we applied a 10% burnin and extracted the Maximum Clade Credibility tree with branch lengths equal to the median of those of all trees using Tree Annotator 1.7.5[53]. Because in preliminary runs Papilionoidea monophyly was not recovered (basal relationships were poorly resolved), we enforced monophyly for this group. The MCC tree where Papilionoidea monophyly was enforced was well resolved, with all Bayesian posterior probabilities > 0.99 (Supplementary Fig. 13).

To assess the robustness of our results to the use of diversity metrics weighted by species abundances we also calculated the species richness as the total number of species seen during the survey period, and the corresponding Chao index[54] to account for imperfect detections (Supplementary Fig. 14) with R package "vegan"[55]. Similarly, we calculated the MPD not accounting for species abundances. For similar reason as for the weighted diversity metric, analysis were performed with the residuals of MPD against species richness or Chao index. The use of either species richness estimates coupled with unweighted MPD residuals did not change qualitatively the results from the ones obtained using the Shannon diversity index and the residual weighted MPD (Supplementary Figs. 7–9). We present in the main text the analysis using the diversity metrics weighted by species abundances.

**Temporal stability and asynchrony measures.** To investigate the mechanisms by which habitat degradation and community diversity might affect the stability of local communities, we calculated the temporal stability of each of these communities as the inverse of the coefficient of variation of the community abundance across time. The temporal stability of a community abundance decreases when the coefficient of variation of the community abundance increases. To quantify the respective contribution of population asynchrony and stability to community stability, we followed Thibaut and Connolly[22] and partitioned the coefficient of variation of community abundance CV into the product of an index of population synchrony φ developed by Loreau and de Mazancourt[56] and the mean coefficient of variation of the population abundance (mean population variability) weighted by

their relative abundances $\overline{CV_w}$ as:

$$CV = \overline{CV_w} \times \sqrt{(\varphi)}. \tag{1}$$

with

$$\overline{CV_w} = \sum_i \frac{\mu_i}{\mu} \frac{\sigma_i}{\mu_i} = \sum_i \frac{\mu_i}{\mu} CV_i, \tag{2}$$

and

$$\varphi = \frac{\sigma^2}{\left(\sum_i \sigma_i\right)^2}. \tag{3}$$

With $\sigma^2$ representing the variance of the abundance of the community, $\sigma_i$ the standard deviation of the abundance of the population $i$ in the community, $\mu$ the temporal mean of the abundance of the community, $\mu_i$ the temporal mean of the abundance of the population $i$ and $CV_i$ the coefficient of variation of the abundance of population $i$. The population synchrony index $\varphi$ ranges from 0 (maximum asynchrony) to 1 (maximum synchrony). To get an asynchrony index that increases with population asynchrony, and population and community indices that increase with population and community stability, respectively, we used the inverses of the synchrony index and of the weighted mean coefficient of variation of population abundance. To achieve normality, we log-transformed the coefficient of variation of community, the coefficient of variation of population and the synchrony index for the three taxa.

We always have

Community stability = 1/2 population asynchrony + weighted mean population stability, (4)

where

$$\text{Community stability} = -\log(CV), \tag{5}$$

$$\text{Population asynchrony} = -\log(\varphi), \tag{6}$$

$$\text{Weighted mean population stability} = -\log(\overline{CV_w}). \tag{7}$$

**Assessing habitat degradation.** To characterize and quantify habitat degradation levels around the monitoring sites, we first used Corine Land Cover [https://www.data.gouv.fr/fr/datasets/corine-land-cover-occupation-des-sols-en-france/] to quantify the percentages of cover occupied by five land-use categories within buffers surrounding the study sites: urban, cropland, heterogeneous agriculture, woodland and seminatural open areas (Supplementary Table 5). To calculate the percentage of cover associated to each of our landscape variables around each study site, we used 3 buffer sizes per taxa: buffers of radius 250, 500, and 1000 m around the transect for bat and around the garden for butterfly communities, and squares of 2, 2.5, and 3 km of side for bird communities. These differences in size and shape among taxa accommodate for the shape of the sampled area and the scale at which landscape is known to affect those taxa[44,57,58]. The use of either of the three buffer sizes in our statistical analyses (see below) did not change qualitatively our results (Supplementary Figs. 7–9).

Second, to account for landscape complexity, we calculated a Shannon diversity index on the area of each land-use category of the level 3 of Corine land cover (Supplementary Table 5).

To account for potential changes in the landscape during the monitoring period, we averaged the percentage areas and landscape complexity described above over the years available in the Corine land cover database that match the monitoring periods. For bird communities, we used the land cover data for the years 2000, 2006, 2012, and 2018; for bat communities, the years 2006 and 2012; and for butterfly communities, the years 2006, 2012, and 2018.

Third, to account for the intensity of both urban and agricultural land uses, we calculated two indices. The area of sealed soil from the European Soil Sealing V2 [http://www.eea.europa.eu/data-and-maps/explore-interactive-maps/european-soil-sealing-v2] (hereafter, sealed soil) that is only available for the year 2006, and an index of agricultural practice intensity (hereafter, agricultural inputs) following the European Union agri-environmental indicator of intensification-extensification [http://ec.europa.eu/eurostat/statistics-explained/index.php/Agri-environmental_indicator_-_intensification_-_extensification]. This last indicator is defined as the sum of expenses in k€ for fertilizers, pesticides, livestock food and veterinarian medics per year divided by the area of agricultural land. It was calculated for each year and administrative region using data from the Agricultural Network of Account Information (RICA) [http://agreste.agriculture.gouv.fr/enquetes/reseau-d-information-comptable/]. This indicator is available for all years except 2017 and 2018. For each site, we calculated the mean over the monitoring period.

We then performed a principal component analysis (PCA) on these eight landscape variables calculated for each monitored site, followed by a Varimax rotation[59] to ease the interpretation. The two first dimensions were used to characterize habitat degradation by an urban gradient and an agricultural intensity gradient, the two being independent from each other (Fig. 2). To achieve normality for the following analysis, we log-transformed the urban gradient for the three taxa.

The PCA was performed twice, once for the Dataset 1 and 2 and once for the Dataset 3 (Supplementary Fig. 15) as the sites included differ between both.

**Statistical analysis**. To assess the relationship between habitat degradation gradients and community stability, we first fitted for each taxonomic group (1) a linear model with the two habitat degradation gradient as explanatory variables (Fig. 3a, b), (2) a linear model with the species diversity and the phylogenetic diversity as explanatory variables (Fig. 3c, d).

To quantify the direct and indirect effects of habitat degradation gradients on community stability, SEM was performed in the following two steps:

First, for each taxonomic group and related buffer sizes, we built a set of five linear models to assess the effects of urban and agricultural intensity gradients on species diversity (model 1) and phylogenetic diversity (model 2), the effects of urban gradient, agricultural intensity gradient, species diversity and phylogenetic diversity on population stability (model 3) and asynchrony (model 4), and the effects of the population stability and asynchrony on community stability (model 5).

Second, to disentangle direct and indirect relationships among variables, and compare the strength of significant relationships, we conducted piecewise SEM[60] for each taxonomic group and related buffer sizes. We used Shipley's test of d-separation to assess the overall fit of the SEM[61]. The strength of a path from a variable on another is the product of the strength of each significant relationship along the path. The overall effect of a variable on another one is the sum of the paths joining the two variables.

For each taxon, the analysis presented in the main text correspond to that performed with the buffer size leading to the lowest AIC. All results remain qualitatively similar for all scales (Supplementary Figs. 7–9).

For all statistical analyses, spatial autocorrelation was accounted for. For bird and butterfly communities we used generalized least squares with exponential and gaussian spatial correlation structures respectively[62]. Because bat communities were aggregated in two distinct geographical regions, we used linear mixed-models with region as random effect (Ile-de-France or Manche)[62].

To further our understanding of the effect of habitat degradation on population stability, we estimated the mean-variance scaling for each taxon by performing a linear mixed-model explaining the log of the variance of population abundances by the log of the mean population abundances, with both species and site as a random effect. We found a slope of $1.61 \pm 0.029$ for bats, $1.09 \pm 0.004$ for birds, and $1.09 \pm 0.018$ for butterflies, meaning that the stability of communities increases with the mean population abundance of the communities[22]. We verified these predictions by looking at the relationships between the weighted mean population stability and the mean population abundance for each taxon using linear models identical to those used in the path analyses (Supplementary Fig. 5). In addition, we tested whether habitat degradation gradients were correlated to either the mean population abundance of the communities or its standard deviation (Supplementary Fig. 4). Statistical analyses were performed using the R software[63], with libraries "PiecewiseSEM v1.2.0"[60], "nlme"[64], "picante"[65], "ape"[66], "ade4"[67], and "lmerTest"[68].

**Reporting summary**. Further information on research design is available in the Nature Research Reporting Summary linked to this article.

## Data availability
The datasets that support the findings of this study are available in Zenodo with the identifier [https://doi.org/10.5281/zenodo.3736101]. Community raw data come from citizen science programs hosted by the Vigie Nature program [http://vigienature.mnhn.fr/]. For bats we used the French bat-monitoring program [http://www.vigienature.fr/fr/chauves-souris]. For birds we used the French Breeding Bird Survey [http://www.vigie-plume.fr/]. For butterflies we used the French garden butterfly observatory [http://www.vigienature.fr/fr/operation-papillons]. Data used to compute land-use areas and landscape complexity are available on the Corine Land Cover website [https://www.data.gouv.fr/fr/datasets/corine-land-cover-occupation-des-sols-en-france/]. Data used to compute sealed soil area are available on the EEA website [http://www.eea.europa.eu/data-and-maps/explore-interactive-maps/european-soil-sealing-v2]. Data used to compute the agricultural inputs are available on the AGRESTE website [http://agreste.agriculture.gouv.fr/enquetes/reseau-d-information-comptable/].

## Code availability
All codes used during the current study are available in Zenodo with the identifier [https://doi.org/10.5281/zenodo.3736101].

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

## Acknowledgements
We thank Frédéric Jiguet, Christian Kerbiriou, and Jean-François Julien for sharing their knowledge on birds and bats; Claire de Mazancourt, Martin Jeanmougin, Emmanuelle Porcher, and Sonia Kéfi for insightful discussions and Alexandre Robert for drawing the animal motifs. Most importantly, we thank all the volunteers who contributed to the citizen science programs STOC-EPS, Vigie-Chiro, and Opération Papillons, run by Vigie-Nature, Ligue pour la Protection des Oiseaux, and Noé Conservation. T.O. was supported with a fellowship by the Chaire "Modélisation Mathématique et Biodiversité" of Veolia Environnement—École Polytechnique—Museum national d'Histoire naturelle—Fondation X, E.T. and C.F. were funded by the ANR project ECOSTAB (ANR-17-CE32-0002/ECOSTAB).

## Author contributions
T.O., E.T., M.E., B.F., and C.F. designed the study and analyzed the results; T.O. managed the datasets; M.E. built the butterfly and the bird phylogenetic trees; T.O., E.T., M.E., B.F., and C.F. participated in writing.

## Competing interests
The authors declare no competing interests.
