## [Peer Review File · Nature Communications]

Peer Review File -

Reviewers' comments:

Reviewer #1 (Remarks to the Author):

This paper examines data from three citizen science projects (birds, bats, butterflies), to see whether species diversity (within taxonomic groups) and habitat degradation (conversion of surrounding habitat) affect community stability (within same taxonomic groups). Similar research has been carried out with plant communities and the authors consider the lack of such studies on animal communities to be an important research gap to fill. The authors use structural equation modeling to tease apart pathways affecting community dynamics. They report that species diversity affects the synchrony of populations and habitat degradation (conversion of surrounding landscapes to agriculture and urban areas) affects the stability of populations.

Overall the findings are interesting and an important contribution to understanding large-scale patterns in community dynamics. I encourage the authors to provide more clarity in terms in order to speak to a broader readership. A table of definitions (ecological and statistical) would be helpful.

The paper could be improved by drawing more on natural histories of the species studied to inform analyses.

I appreciate that the authors were thorough in analyzing data with different landscape buffer sizes, but the landcover data was from 2006 and the agricultural intensity data from 2006-2013, and I wonder how failing to meet assumptions of static landscapes affects the study. The authors could also make it more clear the composition of the focal habitats for each species community.

The citizen science projects are exemplary programs that provide reliable data. I'm concerned about the use of BBS data for the purposes of the paper because the BBS protocol involves fixed dates over years and the data were largely influenced by fall and spring migrants which may have shifted the timing of their migration in response to climate change.

For a paper examining mechanisms of community dynamics, I find it a limitation that communities were examined within taxonomic silos. Species depend on species in other taxonomic groups, so why would we expect the diversity of one taxonomic group to be of high importance in affecting the stability of that same taxonomic group? Related, the authors are vague on the focal habitat type for the taxonomic groups studied --- do they all share the same habitat?

A confusing point for me relates to how (or when) the surrounding landscape was measured. The authors use landcover data from 2006. I've never been to France and so I don't know if the landscape is no longer being converted into new landuses, but a static landscape seems an unlikely assumption. Could the landscape really not have changed over the years of the study (6, 11, 17 years)? The authors also use a value related to the mean agricultural spending between 2006-2013 as index of agricultural intensification, though citizen science data on species abundance were collected through 2018. With the landscape data, the authors measure habitat conversion around focal sites and make implicit assumption about habitat conversion leading to habitat degradation. They should make those assumptions explicit, or confine inferences to the effects of the surrounding habitat conversion (which is related to habitat loss, connectivity, and other landscape features in addition to potential degradation of habitat that remains). Specifically, they assume urbanization is a detriment to semi-natural and agricultural areas while agricultural intensification is a detriment to woodland and semi-natural areas. Why wouldn't agricultural intensification, like urbanization, also be a detriment to agricultural areas? While at least some details are provided about the buffer habitats, I could not find details about the focal habitats. For example, for the bat samples, how many local pops were in each of the different land cover types? For bird samples, how many BBS plots were distributed in which habitat types? Ditto for butterfly garden counts. For a paper examining mechanistic aspects of community dynamics, the authors are vague on natural histories of the taxon studied, and those natural histories are necessary to

make sense of their results. How would it affect the results if some butterfly species had metapopulation dynamics in which "stability" manifests as localized boom-bust dynamics followed by re-colonization events?

With regard to the citizen science projects, the three selected have standardized protocols and long-term data which are two features that favor data quality. As the authors point out, the bat survey is biased for roads with low traffic. I could not find information about the focal habitat for the 152 local bat communities studied. The project requires quite a bit of work from volunteers which leads me to strongly suspect the competent volunteers remain and provide reliable and trustworthy data.

The Birding Bird Survey is 2x2km² plots randomly selected in which 10 survey points to cover range of habitat types within a given plot (but again, not clear details about the proportions of these habitats in the final sample). The BBS involves sampling twice per year on exactly the same dates per site per year, during fall and spring migration which means that these data do not represent the local community of birds, but migrants passing through. Again I suspect this intense project attracts highly competent birders capable of providing reliable and trustworthy data. However, the timing of avian migration peaks have been found to change in response to climate change in some areas, and with a large component of abundance being migrants, it raises question about whether trends in abundance reflect local changes or changes in timing of migration for these 269 bird communities – which they call "local bird communities" but not clear to me whether they excluded migrants.

The Butterfly data come from "gardens" which are residential habitat. The authors are appropriately conservative in restricting the butterfly/month species because identification by volunteers for some is difficult.

Reviewer #2 (Remarks to the Author):

The major claims of this paper is that human land use and species diversity have similar effects on the destabilization of animal communities. The relationship between human impacts on diversity and community stability has been a topic of study for decades in ecology. As such, this paper is on an important topic. However, what the authors present is not groundbreaking. They do look at stability and diversity in a couple of ways, but there is no mechanism. It is a nice empirical study, but I do not know what more it provides than, for example, Blüthgen, Nico, et al. "Land use imperils plant and animal community stability through changes in asynchrony rather than diversity." *Nature Communications* 7 (2016): 10697.

Comments

1. The authors performed a regression on of weighted MPD on SR. Why was a phylogenetic metric that includes species abundances chosen? What is the relationship between unweighted MPD and community stability? Including abundances into an explanatory variable introduces statistical independence biases (i.e., including an independent variable that derived from the dependent variable). The residuals (see figure S15) may certainly help reduce independence biases but then one must assume that the regressions that produced the MPD residuals well fit the data. If the authors want to use the weighted metric, then instead of richness they should use a diversity metric instead. Regardless of choice, the authors must include the unweighted MPD metric as this is more analogous to the SR metric and is the metric most often used. Note that the expectation of unweighted MPD is not dependent on SR (the actual values are but the expectation is not), thus it is less likely that the authors would need to

2. How were the species in the analyses chosen? The authors indicate they only used common species, but where is the cutoff and does this cutoff influence the metrics of community stability? Are the more diverse communities more likely to have rare species that were removed from the analyses and does including these rare species influence the relationships?

3. Statistically, as SR decreases CV will increase and thus, SR should have a negative relationship with community stability as measured by $1/CV$. How then do the authors account for the statistical artifact of lower SR along urbanization and agriculture gradients? Figure 2 does not deal with this issue and the SEM may not as well. I say may not because SEMs are still prone to collinearity issues like any standard regression, but the effect of collinearity must be assessed for any model (it does not seem as if the authors did this check). An often used method is to devise a randomization and thus model (and plot) standardized effect sizes.

4. Comments on figures and tables: Table 1 is not a visually useful presentation of the results. Consider bar charts with errors so show the difference in magnitude. Figure 1: the magnitude of total abundance is so large that you cannot see how the individual species vary. Try plotting two y axes, log transform, or broken y axis. Figure 2: why are only some of the relationships plotted here? The x axis in c is difficult to see the bat data. Perhaps try plotting z scores of richness for each of the data sets.

Reviewer #3 (Remarks to the Author):

This paper is using citizen science biodiversity monitoring data to assess how habitat degradation and urbanization influence the stability of ecological communities, and the relative roles of decreasing population stability and synchrony on community stability. I appreciate the innovative approach, but have some serious doubts both about the data used and regarding the interpretation of the results.

The authors are using monitoring data on birds, bats and butterflies collected by citizen scientists to assess community stability over time. However, the time series for bats is quite short for this types of analyses. According to the authors (lines 290-291) the bat data consist of time series on only 4-6 years. This is very short in an ecological context, and is unlikely to include the full variability of bat communities. Hence, it is likely that they are under-estimating the community variability.

The butterfly data does not consists of surveys of the full community. Instead, citizen scientists are recording species on a "closed list of 28 common species or species groups" (line 278), of which only 14 were used in the analyses. This is probably far from the full set of species encountered over a period of 7-11 years, even in urban environments. The authors state that "the temporal variability of the butterfly community restricted to these 14 species reflects that of the entire community" (lines 282-283). It is however unclear if they by "the entire community" here actually refer to the 28 common "species or species groups", or if they actually refer to the full community. Since they probably only have data on the 28 species or species groups I assume that this is what they refer to. If so, I find it misleading to say that "the temporal variability of the butterfly community restricted to these 14 species reflects that of the entire community". And more importantly, since they do only sample a subset consisting of common species, I find it likely that they are under-estimating the community variability.

For these reasons, there is a considerable risk that the authors are under-estimating the variability of both bat and butterfly communities, with unknown consequences for the main conclusions.

Further comments on the data:

- Butterflies were monitored in residential gardens. Yet most of the monitored species are unlikely to actually breed in many of these gardens, but rather use them for foraging only. So, if only a part of their habitat is monitored it is unclear what the inter-annual variability in community composition actually means.
- Also, the monitored species include two migratory species (*Vanessa atalanta* and *V. cardui*), of which at least *V. cardui* is unlikely to overwinter in France, and the population variability depends mainly on conditions in North Africa and the Mediterranean region.

Other major comments:

In the abstract, the authors state that both population synchrony and population stability have effects on community stability, but through different drivers, i.e. are important, but that diversity loss is mainly affecting synchrony and habitat degradation mainly decreases population stability.

Yet, in the main text the authors mainly discuss the effects of habitat degradation on stability, and very little about diversity and synchrony. This inconsistency needs to be resolved.

The different environmental drivers are estimated at different spatial scales. Landscape variables are assessed within radii of 250, 500 and 1000 m for bird and butterflies and in squares of 2-3 km for birds. In contrast, agricultural intensity variables are assessed with administrative regions (of unclear size, but probably much larger than the areas for which the landscape variables are assessed). This means that it is problematic to conclude that "while butterfly community stability is mainly impacted by urbanization, bird and bat communities are mostly destabilized by agricultural intensification...". In reality, it might instead be that butterfly communities are impacted by (any) drivers at smaller spatial scales while bird and bat communities are impacted by (any) drivers at larger spatial scales.

Also, the radii of 250-1000 m seems quite small for butterflies and especially for bats, which are flying over rather large

Minor comments:

- I would like to see something about methods or the type of data used in the Abstract.

- Lines 285-286: "...gardens that had been monitored in July..." Does this mean that only data from July was included in the analyses? But several of the butterfly species are active mainly during other parts of the year??

- Line 377: Why did you use Corine Land cover data from 2006, and not more recent data (which I believe should be available).

**Reviewer #1:**

**This paper examines data from three citizen science projects (birds, bats, butterflies), to see**
**whether species diversity (within taxonomic groups) and habitat degradation (conversion of**
**surrounding habitat) affect community stability (within same taxonomic groups). Similar**
**research has been carried out with plant communities and the authors consider the lack of such**
**studies on animal communities to be an important research gap to fill. The author use structural**
**equation modeling to tease apart pathways affecting community dynamics. They report that**
**species diversity affects the synchrony of populations and habitat degradation (conversion of**
**surrounding landscapes to agriculture and urban areas) affects the stability of populations.**

**#1. Overall the findings are interesting and an important contribution to understanding large-**
**scale patterns in community dynamics. I encourage the authors to provide more clarity in terms**
**in order to speak to a broader readership. A table of definitions (ecological and statistical) would**
**be helpful.**

We added a supplementary table with ecological and statistical definitions in the manuscript
(supplementary Table 2):

	Definition	Equation
Temporal stability of community total abundance	Reflects the amplitude of inter-annual variations in the total abundance of the community in a site.	$\frac{1}{CV} = \frac{\mu}{\sigma}$
Weighted mean population stability	Mean amplitude of inter-annual variations of population abundances in a community, weighted by their relative abundance.	$\frac{1}{CV_w} = \sum_i \frac{\mu_i \mu_i}{\mu_i \sigma_i}$
Population asynchrony	Reflects the negative correlation degree among species temporal abundance fluctuations in a community.	$\frac{1}{\varphi} = \frac{(\sum_i \sigma_i)^2}{\sigma^2}$
Shannon diversity	Species diversity measure that takes species richness and evenness into account. In this study, we computed for each community the exponential of the Shannon index based on the total community species richness and evenness.	$H' = - \sum_{n=1}^n (p_i * \ln(p_i))$
Species richness	Total number of species seen at least one year during the time series. Species richness is computed for each site and each dataset.	
Weighted or non-weighted Mean Pairwise Distance (MPD)	Reflects the phylogenetic diversity inside a community. The standard MPD is computed as the mean pairwise phylogenetic distance between species in a community. The weighted MPD is computed the same way, but distances are weighted by the product of the abundances of the species that constitute the pair	

Footnote: With CV the coefficient of variation of community abundances, μ the community
mean abundance, σ the standard deviation of community abundance, CV_w the weighted mean
coefficient of variation of population abundance in a community, μ_i the mean abundance of
the population i in a community and σ_i its standard deviation, φ the synchrony between
population abundance fluctuation in a community, H' the Shannon index, n the species of a
community and p_i the proportion of the abundance of the species i in its community.

**#2. The paper could be improved by drawing more on natural histories of the species studied to**
**inform analyses.**

We agree that the natural history of the species studied can inform our analysis and help to interpret
the results. However, we also think that the main strength of the paper is to show common patterns

among taxa with very different natural history and that highlighting too much the peculiarities of each
taxa may blur the main message. To account for this point, we made the following changes in the main
text:

Line 57-61

“While bats and insectivorous birds are recognized as important for pest control^{17,18}, butterflies
contribute to pollination¹⁹ and frugivorous birds are essential for plant dispersal¹⁸. As such,
understanding what determine the stability of these communities might be relevant to understand the
stability of the functions and services they provide.”

Lines 112-116

“This extends classical results found for experimental plant communities^{4,5} to animal taxa with very
different natural history and there by suggest that the positive biodiversity-stability relationship found
for primary production apply to other functions and services provided by various animal
communities.”

**#3. I appreciate that the authors were thorough in analyzing data with different landscape**
**buffer sizes, but the landcover data was from 2006 and the agricultural intensity data from**
**2006-2013, and I wonder how failing to meet assumptions of static landscapes affects the study.**

The reviewer raises an important point. We agree that while the landscape context, i.e., the amount of
degraded land around the study sites, can affect the stability of animal communities, its variation
during the monitoring period could also have an effect.

To assess the extent to which landscape modification occurred in our monitored sites throughout the
study periods, we looked at changes in the land cover using the Corine Land Cover database that is
available for the years 2000, 2006, 2012 and 2018. Using the same buffers surrounding the monitoring
sites as in the main analysis, we quantified the percentage of area that change for one land-use to
another between the years 2000 and 2018 for the birds communities, 2006 and 2012 for the bat
communities, 2006 and 2018 for the butterfly communities. We chose these periods from the available
data to be the closest to the monitoring period for each taxonomic group (see figure R1 below).

Figure R1: Distributions of the amount of landscape surrounding the study sites that have changed
during the study period for each studied taxa.

Our analysis indicates that more than 99%, 76% and 86% of the monitoring sites experience less than
10% change in the landscape surrounding them, for bats, birds and butterflies, respectively. Therefore,
for a large majority of our sites, the hypothesis of static landscape around our study site does not seem
inadequate.

However, to follow the reviewer's concerns on the time-period covered by our landscape variables, we
modified our analyses to better reflect the landscape characteristics around our study sites over the
whole period of monitoring. For bird communities, as time series span from 2001 to 2017, we
measured the average percentages of cover occupied by the five land-use from Corine Land Cover
database over the years 2000, 2006, 2012 and 2018. We also updated the agricultural inputs variable
and used the average of agricultural inputs between 2001 and 2016 (data is not available for the years
2017 and 2018). Similarly, for bats, as time series span from 2006 to 2012, we used the average
percentages of cover from the Corine Land Cover databases for the years 2006 and 2012, as well as
the agricultural inputs average over the years 2006 to 2012. For butterflies, which time series span
from 2007 to 2018, we used the average percentages of cover for the Corine Land Cover databases for
the years 2006, 2012 and 2018 as well as the average of agricultural inputs between 2007 and 2016.
We now present these new analyses in the main text and in the supplementary materials. Please note
that our results were mostly unchanged by these modifications of the landscape variables in our
analysis (see Table below comparing the results of the analyses now presented in the main text with
similar analyses but with previously used landscape variables).

Taxa	Response	Predictor	Main analyses with updated landscape variables			Same analyses with previous landscape variables		
			Est.	SE	P-val	Est.	SE	P-val
Birds	Species diversity	Agricultural intensification gradient	-0.187	0.063	0.003	-0.312	0.074	0.000
	Species diversity	Urbanization gradient	-0.163	0.058	0.006	-0.177	0.055	0.002
	Phylogenetic diversity	Agricultural intensification gradient	0.178	0.066	0.007	0.210	0.076	0.006
	Phylogenetic diversity	Urbanization gradient	-0.016	0.064	0.799	-0.008	0.063	0.905
	Population stability	Agricultural intensification gradient	-0.397	0.065	0.000	-0.419	0.082	0.000
	Population stability	Urbanization gradient	0.081	0.060	0.179	0.041	0.059	0.491
	Population stability	Phylogenetic diversity	0.041	0.056	0.462	0.035	0.057	0.533
	Population stability	Species diversity	-0.017	0.062	0.786	-0.036	0.064	0.577
	Population asynchrony	Agricultural intensification gradient	0.017	0.061	0.784	0.049	0.062	0.429
	Population asynchrony	Urbanization gradient	0.050	0.059	0.399	0.049	0.060	0.419
	Population asynchrony	Phylogenetic diversity	0.030	0.059	0.607	0.024	0.059	0.684
	Population asynchrony	Species diversity	0.366	0.060	0.000	0.377	0.062	0.000
	Community stability	Population asynchrony	0.722	0.000	0.000	0.722	0.000	0.000
	Community stability	Population stability	0.500	0.000	0.000	0.500	0.000	0.000
Butterflies	Species diversity	Agricultural intensification gradient	-0.131	0.082	0.111	-0.152	0.083	0.071
	Species diversity	Urbanization gradient	-0.366	0.082	0.000	-0.314	0.083	0.000
	Phylogenetic diversity	Agricultural intensification gradient	-0.059	0.064	0.358	-0.035	0.069	0.616
	Phylogenetic diversity	Urbanization gradient	0.258	0.061	0.000	0.263	0.061	0.000
	Population stability	Agricultural intensification gradient	0.047	0.083	0.575	0.030	0.085	0.727
	Population stability	Urbanization gradient	-0.366	0.089	0.000	-0.389	0.087	0.000
	Population stability	Phylogenetic diversity	0.202	0.084	0.019	0.207	0.086	0.018
	Population stability	Species diversity	0.133	0.087	0.131	0.140	0.085	0.104
	Population asynchrony	Agricultural intensification gradient	0.176	0.086	0.042	0.126	0.090	0.163
	Population asynchrony	Urbanization gradient	0.102	0.091	0.268	0.086	0.091	0.347
	Population asynchrony	Phylogenetic diversity	0.257	0.087	0.004	0.258	0.090	0.005
	Population asynchrony	Species diversity	0.283	0.090	0.002	0.271	0.090	0.003
	Community stability	Population asynchrony	0.637	0.000	0.000	0.637	0.000	0.000
	Community stability	Population stability	0.710	0.000	0.000	0.710	0.000	0.000

Bats	Species diversity	Agricultural intensification gradient	0.097	0.091	0.290	0.202	0.080	0.012
	Species diversity	Urbanization gradient	0.283	0.085	0.001	0.313	0.080	0.000
	Phylogenetic diversity	Agricultural intensification gradient	-0.192	0.087	0.030	-0.416	0.077	0.000
	Phylogenetic diversity	Urbanization gradient	-0.188	0.080	0.020	-0.239	0.077	0.002
	Population stability	Agricultural intensification gradient	-0.254	0.091	0.006	-0.226	0.091	0.014
	Population stability	Urbanization gradient	-0.105	0.090	0.244	-0.097	0.088	0.270
	Population stability	Phylogenetic diversity	0.059	0.082	0.476	0.037	0.087	0.668
	Population stability	Species diversity	-0.192	0.081	0.019	-0.175	0.083	0.037
	Population asynchrony	Agricultural intensification gradient	0.088	0.086	0.308	0.078	0.085	0.363
	Population asynchrony	Urbanization gradient	-0.039	0.085	0.648	-0.045	0.083	0.588
	Population asynchrony	Phylogenetic diversity	0.243	0.077	0.002	0.243	0.082	0.003
	Population asynchrony	Species diversity	0.391	0.076	0.000	0.390	0.078	0.000
	Community stability	Population asynchrony	0.633	0.000	0.000	0.633	0.000	0.000
	Community stability	Population stability	0.687	0.000	0.000	0.687	0.000	0.000

#4. The authors could also make it more clear the composition of the focal habitats for each species community.

Except for the butterfly monitoring scheme, the monitoring programs do not target specific habitats. Instead, observers were given random locations where to monitor bats or birds. The idea being that all habitats and landscape contexts should be monitored and random locations should lead to sample landscape and habitat proportionally to their area in a given region. Because of this sampling plan design, there is no focal habitat for bird and bat communities: all habitats were sampled proportionally to their surface. For the butterfly monitoring scheme, only private garden were monitored, these gardens being surrounded by various types of landscapes, from urban to rural. For all three taxa, the presence of potential habitats at monitoring sites is characterized by the landscape data that are summarized in the two first axes of the principal component analysis.

To clarify this point, we added the following sentences to the description of the protocols:

For bats:

“These circuits were chosen to be close of the volunteer residency, with low-traffic roads for security and representative of the different land-cover types in the area.”

For birds:

“Plots are squares of 2x2km² randomly selected by the national coordinator, within which the surveyor places 10 points separated by at least 300m, in order to cover all the habitats present in the plot.”

For butterflies:

“Participants identify and count Lepidoptera in their own garden.”

See also our response to comment # 8 of reviewer 1 where we clarified the way landscape context is analysed.

#5. The citizen science projects are exemplary programs that provide reliable data. I'm concerned about the use of BBS data for the purposes of the paper because the BBS protocol involves fixed dates over years and the data were largely influenced by fall and spring migrants which may have shifted the timing of their migration in response to climate change.

The French Breeding Bird Survey program is based on records only during the spring period and not during the fall period, avoiding an influence of fall migrants. Moreover, the two sessions required in this program have to be done over a 3-month period (from April 1st to June 30th), one before 8 May, the second one after, and have to be separated by at least 4 weeks. For a given site, survey dates have to be the same over the years, with a tolerance of a +/- 5 days, to allow for observer availability (most

127 observers are volunteers who participate on their spare time) and weather conditions. This information
is present in the description of the bird monitoring program in the method section (lines 413-418).

Previous studies found that bird migrants arrived at their breeding grounds on average 2 days per
decade earlier (Bitterlin and Buskirk, 2014; Gienapp et al., 2007; Lehikoinen et al., 2004; Rubolini et
al., 2007; Usui et al., 2017). Over the seventeen years of the monitoring period used in this study
(2001-2018) the shift of migrating dates due to climate change is therefore less than four days, i.e. less
than the shift of monitoring dates allowed by the protocol. It seems thus unlikely that shifts in arrival
dates bias our data.

**#6. For a paper examining mechanisms of community dynamics, I find it a limitation that**
**communities were examined within taxonomic silos. Species depend on species in other**
**taxonomic groups, so why would we expect the diversity of one taxonomic group to be of high**
**importance in affecting the stability of that same taxonomic group? Related, the authors are**
**vague on the focal habitat type for the taxonomic groups studied --- do they all share the same**
**habitat?**

We agree with the reviewer that analysing the three communities together would be excellent.
However, the data comes from three different citizen science programs; each one focalised on a given
taxonomic group. As a consequence each site provides data for only one taxonomic group, making it
impossible to analyse the effect of one taxonomic group on the others.

We modified the main text to make sure that readers understand that each site contains only one
taxonomic group counts, by adding the following sentence line 55-57:

“The three taxonomic groups were not monitored on the same sites as data come from three
independent citizen science programs.”

Regarding the focal habitat, see our response to comment # 4 reviewer # 1 and response to comment #
3 reviewer # 3

**#7. A confusing point for me relates to how (or when) the surrounding landscape was measured.**
**The authors use landcover data from 2006. I’ve never been to France and so I don’t know if the**
**landscape is no longer being converted into new landuses, but a static landscape seems an**
**unlikely assumption. Could the landscape really not have changed over the years of the study**
**(6, 11, 17 years)? The authors also use a value related to the mean agricultural spending between**
**2006-2013 as index of agricultural intensification, though citizen science data on species**
**abundance were collected through 2018.**

As outlined in our response to comment #3 of reviewer 1, we now include additional land cover data
(Corine Land Cover for 2000, 2006, 2012 and 2018) as well as additional data on mean agricultural
inputs between 2001 and 2016 in order to better quantify average landscape variables over the whole
monitoring periods of the different communities. We modified the methods in relation to these
changes (lines 425-466) and now explain more precisely how the surrounding landscape was
measured around each monitored community.

Most landscapes surrounding our sampled communities have only changed slightly over the
monitoring periods (see Figure R1 above). Regarding the index of agricultural intensification, we now
consider the mean agricultural spending over the entire monitored period of each taxa respectively.

**#8. With the landscape data, the authors measure habitat conversion around focal sites and**
**make implicit assumption about habitat conversion leading to habitat degradation. They should**
**make those assumptions explicit, or confine inferences to the effects of the surrounding habitat**
**conversion (which is related to habitat loss, connectivity, and other landscape features in**

addition to potential degradation of habitat that remains). Specifically, they assume
urbanization is a detriment to semi-natural and agricultural areas while agricultural
intensification is a detriment to woodland and semi-natural areas. Why wouldn't agricultural
intensification, like urbanization, also be a detriment to agricultural areas? While at least some
details are provided about the buffer habitats, I could not find details about the focal habitats.
For example, for the bat samples, how many local pops were in each of the different land cover
types? For bird samples, how many BBS plots were distributed in which habitat types? Ditto for
butterfly garden counts.

We believe that there is a misunderstanding here. We do not measure habitat conversion but landscape
composition. The two landscape gradients extracted from the principal component analysis (PCA)
indicate that in the set of sites we studied, we have sites with various level of urbanization (first axis of
the PCA) and the more urban area around a site there is, the less semi natural habitat and agricultural
area there are. Independently of this gradient, our monitored sites vary in the amount of intensive
agricultural land surrounding them (second axis of the PCA), from sites surrounded by intensive
agricultural lands to sites surrounded by woodland and/or open semi-natural areas.

To clarify this point, we modified the description of the analysis in the main text line 63 to 70 (see
below) and integrated the supplementary figure 1 in the main text with expended legend. We also now
avoid using the terms "agricultural intensification" and "urbanization" and rather refer to "agricultural
intensity gradient" and "urban gradient".

"We analyzed the landscape surrounding these communities using a principal component analysis and
distinguished two independent habitat degradation gradients. First an urbanization gradient opposing
sites surrounded by urban and sealed soil areas to sites surrounded by semi-natural and agricultural
landscape. Second, an agricultural intensification gradient, opposing sites within landscapes
dominated by cropland areas with high agricultural inputs to sites surrounded by heterogenous
landscape including higher proportion of, woodland areas and semi-natural open areas (Fig. 2, see
Methods)."

Regarding the focal habitat, see our response to comment # 4 reviewer # 1 and response to comment #
3 reviewer # 3

**#10. For a paper examining mechanistic aspects of community dynamics, the authors are vague
on natural histories of the taxon studied, and those natural histories are necessary to make sense
of their results. How would it affect the results if some butterfly species had metapopulation
dynamics in which "stability" manifests as localized boom-bust dynamics followed by re-
colonization events?**

See our response to comment # 2 reviewer # 1 regarding the natural history.

The use of nationwide dataset on natural community, necessarily implying correlative approaches,
prevents us to test for all possible mechanisms. Such dataset are necessarily noisy but still can reveal
general patterns that can be related to what is expected from theory or can generalized results found in
smaller but more controlled systems.

We agree that metapopulation dynamics might be one of the mechanisms underlying the dynamic of
some species included in the datasets. However, we do not think that this should question our results.
If the population of such species exhibit boom-bust dynamics at the monitored sites, they will increase
the community variability through their high population variability. However, this should not prevent
them to also be affected by the quality landscape surrounding the sampling site.

**#11. With regard to the citizen science projects, the three selected have standardized protocols
and long-term data which are two features that favor data quality. As the authors point out, the
bat survey is biased for roads with low traffic. I could not find information about the focal**

**habitat for the 152 local bat communities studied. The project requires quite a bit of work from**
**volunteers which leads me to strongly suspect the competent volunteers remain and provide**
**reliable and trustworthy data.**

Regarding the focal habitat, see our response to comment # 4 reviewer # 1 and response to comment #
3 reviewer # 3

There is indeed some turnover of volunteers in these schemes, some of them participating for one or
two years only. However, we selected only sites which were monitored by the same volunteers for a
longer period, finding a compromise between the number of available sites and the number of years of
monitoring. This selection imply that we selected the most motivated and skilled volunteers which
provide data with the highest quality.

**#12. The Birding Bird Survey is 2x2km2 plots randomly selected in which 10 survey points to**
**cover range of habitat types within a given plot (but again, not clear details about the**
**proportions of these habitats in the final sample). The BBS involves sampling twice per year on**
**exactly the same dates per site per year, during fall and spring migration which means that these**
**data do not represent the local community of birds, but migrants passing through. Again I**
**suspect this intense project attracts highly competent birders capable of providing reliable and**
**trustworthy data. However, the timing of avian migration peaks have been found to change in**
**response to climate change in some areas, and with a large component of abundance being**
**migrants, it raises question about whether trends in abundance reflect local changes or changes**
**in timing of migration for these 269 bird communities – which they call “local bird**
**communities” but not clear to me whether they excluded migrants.**

Contrary to the reviewer remark, the French BBS protocol allows for a few days flexibility for
surveying dates: each year, the surveys are done at approximately the same dates to allow for observer
availability (most observers are volunteers who participate on their spare time) and weather
conditions.

Most of the data collected during the surveys represent local breeders (either from sedentary or
migratory species), since the 5 minutes point counts done in the early morning record mostly singing
birds, i.e. territorial males on their breeding grounds. Consequently, there is a low probability that the
counts include migrants passing through.

The French BBS does not include a survey date during fall migration.

We re-run the analysis with the Dataset 1 after removing the transaharian migrants, with the Shannon
index and the weighted MPD. Our results show the robustness of our analysis (see comparison in the
two tables below).

Table: Strength of the different paths by which diversity and habitat degradation affect the stability of
local communities of birds comparing the results of the analyses now presented in the main text with
similar analyses but without transaharian migrants.

Effects on community stability	Results from main analysis	Results excluding transaharian migrants
Diversity effects		
Total effects	0.264	0.218
Species diversity effects	0.264	0.218
Phylogenetic diversity effects	NA	NA
Effects via population stability	NA	NA
Effects via population asynchrony	0.264	0.218

Habitat degradation effects		
Total effects	-0.291	-0.226
Urbanisation effects	-0.043	0.036
Agricultural intensification effects	-0.248	-0.262
Effects via diversity	-0.092	-0.087
Effects via population stability	-0.198	-0.139
Effects via population asynchrony	NA	NA

Table: Standardized coefficients from the Structural Equation Models on the stability of bird communities for the analyses now presented in the main text and similar analyses but without transaharian migrants.. Est. stands for estimates. SE for standard errors. P-val. for P-value.

Response	Predictor	Results from main analysis			Results excluding transaharian migrants		
		Est,	SE	P-val	Est,	SE	P-val
Species diversity	Agricultural intensity gradient	-0.1867	0.0625	0.003	-0.238	0.063	0.000
Species diversity	Urban gradient	-0.163	0.0585	0.006	-0.162	0.061	0.008
Phylogenetic diversity	Agricultural intensity gradient	0.1779	0.0659	0.007	0.266	0.064	0.000
Phylogenetic diversity	Urban gradient	-0.0164	0.0644	0.799	0.033	0.061	0.594
Population stability	Agricultural intensity gradient	-0.3971	0.0646	0	-0.391	0.065	0.000
Population stability	Urban gradient	0.0808	0.06	0.179	0.132	0.058	0.024
Population stability	Phylogenetic diversity	0.0409	0.0555	0.462	0.058	0.058	0.323
Population stability	Species diversity	-0.0167	0.0617	0.786	0.073	0.059	0.212
Population asynchrony	Agricultural intensity gradient	0.0167	0.0607	0.784	-0.015	0.065	0.819
Population asynchrony	Urban gradient	0.0503	0.0595	0.399	0.018	0.060	0.761
Population asynchrony	Phylogenetic diversity	0.0305	0.0593	0.607	0.069	0.063	0.276
Population asynchrony	Species diversity	0.3663	0.0602	0	0.317	0.062	0.000
Community stability	Population asynchrony	0.7216			0.689		
Community stability	Population stability	0.4996			0.538		

The Butterfly data come from “gardens” which are residential habitat. The authors are appropriately conservative in restricting the butterfly/month species because identification by volunteers for some is difficult.

We thank you for your positive comment.

Reviewer #2:

The major claims of this paper is that human land use and species diversity have similar effects on the destabilization of animal communities. The relationship between human impacts on diversity and community stability has been a topic of study for decades in ecology. As such, this paper is on an important topic. However, what the authors present is not groundbreaking. They do look at stability and diversity in a couple of ways, but there is no mechanism.

The use of nationwide dataset on natural community, necessarily implying correlative approaches, prevents us to test for all possible mechanisms. Such dataset are necessarily noisy but still can reveal

general patterns that can be related to what is expected from theory or can generalized results found in
smaller but more controlled systems.

We disagree with the reviewer regarding the lack of mechanisms. As presented in our study, there are
two pathways by which community stability can be affected, either by a change in the stability of the
species themselves or by a change in the synchrony of their fluctuation through time. These two
pathways can involve different mechanisms driving community stability and, actually, our analyses
show that if habitat degradation and diversity loss both decrease community stability, they do it via
different pathways. While habitat degradation mainly decrease the stability of the species within
communities, diversity loss mainly decrease the asynchrony among species within communities. To
our knowledge, these results were not reported in the literature before; they change the classical view
of perturbation decreasing diversity, which in turn decrease stability. Further, our analysis suggests
that the mechanism driving asynchrony is related to response diversity among species since it correlate
to the phylogenetic diversity of the community. Finally, our results indicates that the type of landscape
degradation that affects the most the stability of the populations differ among taxa and we propose
related mechanisms in our discussion such as, depending on the taxa, lower resources availability or
higher variability in resource availability (see lines 159-173 in the main text), thereby paving the way
to more detailed studies.

**#1. It is a nice empirical study, but I do not know what more it provides than, for example,**
**Blüthgen, Nico, et al. "Land use imperils plant and animal community stability through changes**
**in asynchrony rather than diversity." *Nature Communications* 7 (2016): 10697.**

We thanks the reviewer for his or her positive comment. The main difference between our study and
the one of Blüthgen et al. relies on the model of path analysis used that differ both in terms of the
variables included and in terms of the links among them. This has profound implications in the results
and related interpretation.

Firstly, Blüthgen et al. do not include population stability in their path analysis model despite its
known role in determining community stability (Thibaut & Connolly 2013). Consequently, there is no
results in Blütghen et al. regarding the effect of landscape degradation on population stability whereas
our results identify this pathway as dominant for the three taxonomic group studied.

Secondly, the path model used by Blüthgen et al. tests three pathways or mechanisms by which land
use intensity can affect community stability: the first one through a change in diversity, the second
through a change in asynchrony and the third one through a change in abundance. Looking at
Blüthgen supplementary figures 2 and 3, their results clearly indicate that asynchrony has the strongest
effect of community stability compared to diversity or abundance, for all taxonomic groups studied
and for both woodland and grassland. However, they found that land-use intensity significantly affects
species asynchrony only for birds in forests and for plants in grasslands and the effect is negative only
for birds; so they found that land use intensity affects community stability through a change in
asynchrony in only two cases out of eight, one time negatively and one time positively. They also
found that land use intensity affects community stability through a change in abundance in four cases
out of eight (three times negatively and one time positively) and through a change in diversity in two
cases out of eight (both times negatively). Therefore, contrary to their claim, their result do not show a
consistent effect of land use intensity on community stability through a decrease of asynchrony. The
fact that they did not find such an effect is also coherent with our results as we show that the effect of
habitat degradation is not mediated by a change in asynchrony but a change in population stability, a
variable not included in their path analysis models.

Thirdly, the analytical decomposition of the community stability into a species stability and
asynchrony components (Thibaut & Connolly 2013) supports the four storeys architecture of our path
analysis model as it indicates that asynchrony cannot be positioned at the same level as diversity, as
done in Blüthgen et al. As a consequence of this, the architecture of our path analysis model further
allows testing for the effect of habitat degradation both directly through changes in the two
components of community stability and indirectly through changes in diversity. This is another major
difference between our study and that of Blüthgen, as it allows to properly test whether the effects of

habitat degradation are mediated by changes in diversity or directly affect the components of
community stability.

Finally, unlike Blüthgen's study, our analysis also includes phylogenetic diversity as a possible
determinant of community stability, with result supporting the hypothesis that species response
diversity to environmental variations is a key mechanism for population asynchrony in communities.

Refererence:

Thibaut, L. M. & Connolly, S. R. Understanding diversity–stability relationships: towards a unified
model of portfolio effects. *Ecology Letters* **16**, 140–150 (2013)

**#2. The authors performed a regression on of weighted MPD on SR. Why was a phylogenetic**
**metric that includes species abundances chosen? What is the relationship between unweighted**
**MPD and community stability? Including abundances into an explanatory variable introduces**
**statistical independence biases (i.e., including an independent variable that derived from the**
**dependent variable). The residuals (see figure S15) may certainly help reduce independence**
**biases but then one must assume that the regressions that produced the MPD residuals well fit**
**the data. If the authors want to use the weighted metric, then instead of richness they should use**
**a diversity metric instead. Regardless of choice, the authors must include the unweighted MPD**
**metric as this is more analogous to the SR metric and is the metric most often used. Note that**
**the expectation of unweighted MPD is not dependent on SR (the actual values are but the**
**expectation is not), thus it is less likely that the authors would need to.**

We agree with the reviewer that using analogous diversity metrics that either both include species
abundances or not is preferable. We thus rerun our analysis the two ways, either with the Shannon
diversity and the weighted MPD or with the species richness (or Chao index) and the MPD. Both
analysis gave qualitatively similar results (see supplementary figures 15-17).

We think that accounting for species abundance seems to make more sense. Indeed phylogenetic
diversity is expected to foster community stability through increased asynchrony because unrelated
species are expected to respond to perturbation in different ways because of their trait differences. This
mechanism is expected to be stronger when the abundance of those species are even. Similarly, for
species diversity, in uneven communities, a smaller number of species will drive the stability of the
community because of the mean-variance relationship.

We choose to present the analysis with weighted metrics in the main text and to provide the analysis
with unweighted metrics in the supplementary material but this can change if the reviewer or the editor
prefer the other way.

**#3. How were the species in the analyses chosen? The authors indicate they only used common**
**species, but where is the cutoff and does this cutoff influence the metrics of community stability?**
**Are the more diverse communities more likely to have rare species that were removed from the**
**analyses and does including these rare species influence the relationships?**

Regarding the bat dataset, we did not use cut-off and all contacted species were included.

Regarding the butterfly dataset, we did not use cut-off neither. The species list used was defined by
the monitoring program protocol that focuses on 14 species that are easily identifiable and known to
be common from expert knowledge. As shown in our response to comment # 2 of reviewer # 3, the
stability metric calculated with these 14 species was highly correlated with the one calculated with the
extra species and species groups. Note that we cannot perform our analysis with species groups as it
precludes calculating species richness or phylogenetic diversity. Further, including the extra species is
problematic as monitoring these extra species is not mandatory and we cannot know if volunteers do
not report these species because they do not observe them or because they do not use the list of extra
species, which would make our dataset not homogeneous.

Regarding the bird dataset, the 75 species are indeed among the most common ones, excluding species
 often seen while transiting. This selection of species was performed by the coordinator of the
 monitoring program to insure high quality data and to allow producing reliable population trends
 (Devictor et al. 2012; note in this publication that 65 species were found to have sufficient data and
 since then, data on 10 additional species has been added to this list).
 The reviewer is right about the fact that more rare species were found in the communities with the
 more species in the dataset with 75 species (see below).

 Figure R2: correlation between the bird species richness when including only the 75 most common
 species and the full species richness (all species included).

 To assess the impact of excluding rare species, we re-run the analysis with the full set of species,
 including 234 bird species. The results were qualitatively similar to the ones obtained with the 75 most
 common species (see comparison in the two tables below).

Table: Strength of the different paths by which diversity and habitat degradation affect the stability of
 local communities of birds comparing the results of the analyses presented in the main text for
 common bird species with similar analyses but including rare bird species.

Effects on community stability	Results from main analysis with 75 common bird species	Results including rare bird species (total = 234 species)
Diversity effects		
Total effects	0.264	0.175
Species diversity effects	0.264	0.175
Phylogenetic diversity effects	NA	NA
Effects via population stability	NA	-0.076
Effects via population asynchrony	0.264	0.252
Habitat degradation effects		
Total effects	-0.291	-0.24
Urbanisation effects	-0.043	-0.027
Agricultural intensification effects	-0.248	-0.212
Effects via diversity	-0.092	-0.027
Effects via population stability	-0.198	-0.212
Effects via population asynchrony	NA	NA

Table: Standardized coefficients from the Structural Equation Models on the stability of bird
 communities for the analyses presented in the main text and similar analyses but with including rare
 bird species. Est. stands for estimates. SE for standard errors. P-val. for P-value.

Response	Predictor	Results from main analysis with common bird species			Results including rare bird species		
		Est.	SE	P-val	Est.	SE	P-val
Species diversity	Agricultural intensity gradient	-0.1867	0.0625	0.003	-0.1	0.064	0.09
Species diversity	Urban gradient	-0.163	0.0585	0.006	-0.2	0.062	0.012
Phylogenetic diversity	Agricultural intensity gradient	0.1779	0.0659	0.007	0.3	0.062	0
Phylogenetic diversity	Urban gradient	-0.0164	0.0644	0.799	0.07	0.059	0.238
Population stability	Agricultural intensity gradient	-0.3971	0.0646	0	-0.4	0.065	0
Population stability	Urban gradient	0.0808	0.06	0.179	0.05	0.059	0.414
Population stability	Phylogenetic diversity	0.0409	0.0555	0.462	-0.1	0.06	0.387
Population stability	Species diversity	-0.0167	0.0617	0.786	-0.1	0.057	0.014
Population asynchrony	Agricultural intensity gradient	0.0167	0.0607	0.784	-0	0.063	0.684
Population asynchrony	Urban gradient	0.0503	0.0595	0.399	0.04	0.06	0.514
Population asynchrony	Phylogenetic diversity	0.0305	0.0593	0.607	-0	0.063	0.943
Population asynchrony	Species diversity	0.3663	0.0602	0	0.35	0.059	0
Community stability	Population asynchrony	0.7216			0.71		
Community stability	Population stability	0.4996			0.54		

Reference:

(Jiguet, F., Devictor, V., Julliard, R., & Couvet, D. (2012). French citizens monitoring ordinary birds provide tools for conservation and ecological sciences. *Acta Oecologica*, 44, 58-66.)

**#4. Statistically, as SR decreases CV will increase and thus, SR should have a negative**
 **relationship with community stability as measured by 1/CV. How then do the authors account**
 **for the statistical artifact of lower SR along urbanization and agriculture gradients? Figure 2**
 **does not deal with this issue and the SEM may not as well. I say may not because SEMs are still**
 **prone to collinearity issues like any standard regression, but the effect of collinearity must be**
 **assessed for any model (it does not seem as if the authors did this check). An often used method**
 **is to devise a randomization and thus model (and plot) standardized effect sizes.**

To check for the potential multi-collinearity in our models, we performed variance inflation factor
 analysis for the models presented in the main text, i.e. the Dataset 1, the spatial scale with the best AIC
 and for both Shannon index / weighted MPD and species richness / non-weighted MPD. All our
 variance inflation factor are less than 1.5, suggesting the absence of multi-collinearity in our models.

Table. Variance inflation factors for each variable / model / taxa.

Mod 1: Shannon index (or species richness) ~ urban gradient + agricultural gradient
 Mod 2: Weighted MPD (or Non-weighted MPD) ~ urban gradient + agricultural gradient
 Mod 3: Weighted mean population stability ~ urban gradient + agricultural gradient+ Shannon index
 (or species richness) + Weighted MPD (or Non-weighted MPD)
 Mod 4: Asynchrony ~ urban gradient + agricultural gradient+ Shannon index (or species richness) +
 Weighted MPD (or Non-weighted MPD)

Shannon index + weighted MPD			Species richness + non-weighted MPD		
Bats	Birds	Butterflies	Bats	Birds	Butterflies

	Urban gradient	1.09	1	1	1.09	1	1
Mod 1	Agricultural gradient	1.09	1	1	1.09	1	1
	Urban gradient	1.09	1	1.01	1.09	1.01	1
Mod 2	Agricultural gradient	1.09	1	1.01	1.09	1.01	1
	Urban gradient	1.29	1.04	1.22	1.25	1.06	1.27
	Agricultural gradient	1.44	1.09	1.17	1.43	1.17	1.06
	Shannon index / species richness	1.16	1.09	1.22	1.11	1.03	1.26
Mod 3	Weighted MPD - Non-weighted MPD	1.25	1.01	1.17	1.31	1.18	1.07
	Urban gradient	1.29	1.09	1.22	1.25	1.05	1.27
	Agricultural gradient	1.44	0.17	1.17	0.143	1.2	1.06
	Shannon index / species richness	1.16	1.16	1.22	1.11	1.08	1.26
Mod 4	Weighted MPD - Non-weighted MPD	1.25	1.05	1.17	1.32	1.14	1.07

#5. Comments on figures and tables: Table 1 is not a visually useful presentation of the results. Consider bar charts with errors so show the difference in magnitude. Figure 1: the magnitude of total abundance is so large that you cannot see how the individual species vary. Try plotting two y axes, log transform, or broken y axis. Figure 2: why are only some of the relationships plotted here? The x axis in c is difficult to see the bat data. Perhaps try plotting z scores of richness for each of the data sets.

Although we agree that a table might not be the most efficient visually, we think that presenting those numbers with a graph might not be the most appropriate either. Table 1 summarizes at the path level the effects presented graphically in Figure 4. It allows comparing the various paths for the three studied taxa in, we think, a very efficient and concise way. Consequently, we prefer to stick to a table. However, we took this reviewer idea to present the results of the robustness analysis in the supplementary figures 15, 16 and 17.

For Figure 1, we log transformed y axes.

For Figure 2, which is now Figure 3 in the main text, we plotted all relationships and we plotted z scores of species diversity and phylogenetic diversity to improve readability.

Reviewer #3:

This paper is using citizen science biodiversity monitoring data to assess how habitat degradation and urbanization influence the stability of ecological communities, and the relative roles of decreasing population stability and synchrony on community stability. I appreciate the innovative approach, but have some serious doubts both about the data used and regarding the interpretation of the results.

#1. The authors are using monitoring data on birds, bats and butterflies collected by citizen scientists to assess community stability over time. However, the time series for bats is quite short for this types of analyses. According to the authors (lines 290-291) the bat data consist of time series on only 4-6 years. This is very short in an ecological context, and is unlikely to include the full variability of bat communities. Hence, it is likely that they are under-estimating the community variability.

Note that the objective of our study is to assess the impact of diversity loss and habitat degradation on the community stability and not to quantify the full variability of the studied communities. Since the length of the time series we used does not vary with the diversity of the studied communities nor with

509 any of the two landscape gradients, there is no reason to believe that the length of the time series could
bias our results.

The reviewer does not either suggest that longer time series would affect differently richer
communities or communities surrounded by intensive agricultural land, urban areas or more natural
landscape, which would be a problem.

Finally, we do not agree that the length of our time series are too short to assess community stability
nor that they are particularly short regarding other studies on the same topic. The mean life expectancy
averaged across species included in our analysis is 2.9 years for bats (see Table 1 for species level
estimates and references), which is under our time series lengths. Hence, for each taxon, our time
series are sufficient to capture community stability over several generations. Also note that compared
to classical studies on diversity-stability relationship in plant communities, the ratio of the length of
the time series on the mean annual life span of the organisms we used is rather high (see table 2 in
Gross et al. 2014, average time series length = 6.5 years). For example in the well-known Cedar Creek
biodiversity experiment, all plants are perennials (life span higher than 2 years) and the time series
used were 10-year long (Tilman et al. 2006 in Nature), which conservatively leads a ratio similar to
what we have for the bat dataset. Finally, our analysis clearly shows that our results are robust to the
length of the time series used (see Tables S3-5 of our manuscript), and recent publications (Isbell et al.
2015 in Nature and Hautier et al 2015 in Science) highlight that results on community stability are
consistent between short-term – 3-4 years - and longer term studies.

Table 1. Mean life expectancy in years of bat species included in our dataset

Species	Average life span (years)
Nyctalus leisleri	2.7
Nyctalus noctula	2.2
Eptesicus serotinus	NA
Pipistrellus pipistrellus	2.2
Pipistrellus kuhlii	2.2
Pipistrellus nathusii	2.9
Barbastella barbastellus	5.5

References

Arthur L. & Lemaire M. 2009. Les Chauves-souris de France, Belgique, Luxembourg
et Suisse. Biotope, Mèze (Collection Parthénope); MNHN, Paris, 544p.

Gross, K., Cardinale, B. J., Fox, J. W., Gonzalez, A., Loreau, M., Wayne Polley, H., ... & van
Ruijven, J. (2013). Species richness and the temporal stability of biomass production: a new
analysis of recent biodiversity experiments. *The American Naturalist*, 183(1), 1-12.

Tilman, D., Reich, P. B., & Knops, J. M. (2006). Biodiversity and ecosystem stability in a
decade-long grassland experiment. *Nature*, 441(7093), 629.

**#2. The butterfly data does not consists of surveys of the full community. Instead, citizen**
**scientists are recording species on a “closed list of 28 common species or species groups” (line**
**278), of which only 14 were used in the analyses. This is probably far from the full set of species**
**encountered over a period of 7-11 years, even in urban environments. The authors state that**
**“the temporal variability of the butterfly community restricted to these 14 species reflects that of**
**the entire community” (lines 282-283). It is however unclear if they by “the entire community”**
**here actually refer to the 28 common “species or species groups”, or if they actually refer to the**
**full community. Since they probably only have data on the 28 species or species groups I assume**
**that this is what they refer to. If so, I find it misleading to say that “the temporal variability of**
**the butterfly community restricted to these 14 species reflects that of the entire community”.**

**And more importantly, since they do only sample a subset consisting of common species, I find it**
**likely that they are under-estimating the community variability. For these reasons, there is a**
**considerable risk that the authors are under-estimating the variability of both bat and butterfly**
**communities, with unknown consequences for the main conclusions.**

We agree that the use of “entire community” was misleading and we replaced it by “the community
including the abundance of the species groups”.

To investigate the potential consequences of not sampling the full community on the estimation of the
community stability we used some extra-data. In addition to the 28 species and species groups, the
volunteers can also use an additional list with three species groups and three species of Rhopalocera
and six Heterocera species. However, the use of this extra list is not mandatory and actually few
participant use it, although we don't know if it is because they don't use it or because they do not
observe the corresponding butterflies. We recalculated the community variability with the 28 species
and species groups as well as this extra list, and compared it to the variability we have with the dataset
including only the 14 species. As shown in the figure below, we found a high correlation between
these two datasets, similarly to what we found when comparing the 14 species dataset to the 28
species and group of species dataset (result presented in the supplementary Fig. 9).

Figure R3. Relationship between the coefficients of variation of the total abundance of the butterfly
communities calculated with the 28 species and group of species plus the extra list and the butterfly
communities restricted to the 14 species that could be identified with certainty (Estimate = 0.57, p-
value < 0.001, intercept = 0.11). Regression line and its 95% confidence intervals are represented by
solid and dotted lines.

This high correlation between the three subsets is not surprising as rare species contribute less to the
variability of the community as their abundance is low. Note that contrary to the expectation of the
reviewer, we tend to overestimate community variability by restricting the data to the 14 species. But
as explained in our response to comment # 1 of reviewer # 3, the objective of our study is not to
provide the true value of community variability but to assess how variability is affected by community
diversity and landscape quality.

Since (i) we need species level identification to perform our analysis, (ii) we cannot be sure that all
volunteers use the extra list and (iii) the community variability is highly correlated among the three
datasets, we are confident that using the dataset on the 14 species does not bias the result of our
analysis and insure that our dataset is homogeneous in term of sampling.

Regarding the impact of including rare species on our results, please see our response to comment # 3
of reviewer # 2.

**#3. Butterflies were monitored in residential gardens. Yet most of the monitored species are**
 **unlikely to actually breed in many of these gardens, but rather use than for foraging only. So, if**
 **only a part of their habitat is monitored it is unclear what the inter-annual variability in**
 **community composition actually means.**

All the 14 butterfly species studied forage in gardens but we agree that we do not know if they actually
 breed in it. However, monitoring the abundance of species within a foraging site still provide
 information on the dynamic of these species, even if the monitoring site does not encompass their full
 habitat. If the population decline, less individuals will be seen foraging and if the population increases,
 more individuals will be seen foraging. Such site-based approaches are common to study community
 dynamic under natural condition (e.g. Blüthgen et al. 2015) and even inevitable since populations
 from different species have different habitat definitions and sizes, not fully overlapping, making it
 impossible to define a common area that would include the entire populations of all species composing
 an ecological community.

**#4. Also, the monitored species include two migratory species (*Vanessa atalanta* and *V. cardui*),**
 **of which at least *V. cardui* is unlikely to overwinter in France, and the population variability**
 **depends mainly on conditions in North Africa and the Mediterranean region.**

We agree that the population variability of migratory species depend on conditions that we do not
 account for in our analysis. However, the local conditions we analysed might still play a role by
 affecting the survival of individuals. Further, if migratory species, which population variability are
 driven by conditions outside our study area, were fully driving the dynamic of the studied
 communities, we would expect that to blur the signal and not to find any effect of the landscape
 characteristics surrounding our study sites. So including these migratory species is conservative
 regarding the result we find.

 However, to test the effect of these migratory species on our results, we re-run our analyses removing
 *V. atalanta* and *V. cardui* (see below). The results obtained are qualitatively similar, highlighting the
 same relationships between habitat degradation, community diversity and community stability.
 Exclusion of these two migratory species mainly lead to the detection of stronger negative effects of
 habitat degradation on butterfly community stability, suggesting that observed effects of landscape
 characteristics might indeed be partly attenuated by the presence of migratory species.

Table: Strength of the different paths by which diversity and habitat degradation affect the stability of
 local communities of butterflies comparing the results of the analyses presented in the main text with
 similar analyses but excluding *Vanessa atalanta* and *V. cardui*.

Effects on community stability	Results from main analysis	Results excluding V. atalanta and V. cardui
Diversity effects		
Total effects	0.487	0.413
Species diversity effects	0.18	0.26
Phylogenetic diversity effects	0.307	0.153
Effects via population stability	0.143	NA
Effects via population asynchrony	0.344	0.413
Habitat degradation effects		
Total effects	-0.134	-0.306
Urbanisation effects	-0.247	-0.254
Agricultural intensification effects	0.112	-0.052
Effects via diversity	0.013	-0.105
Effects via population stability	-0.26	-0.201

Effects via population asynchrony	0.112	NA
-------	----

Table: Standardized coefficients from the Structural Equation Models on the stability of butterfly communities for the analyses presented in the main text and similar analyses but with including rare *Vanessa atalanta* and *V. cardui*. Est. stands for estimates. SE for standard errors. P-val. for P-value.

Response	Predictor	Results from main analysis			Results excluding V. atalanta and V. cardui		
		Est.	SE	P-val	Est.	SE	P-val
Species diversity	Agricultural intensity gradient	-0,131	0,082	0,111	-0.200	0.081	0.015
Species diversity	Urban gradient	-0,366	0,082	0,000	-0.353	0.081	0.000
Phylogenetic diversity	Agricultural intensity gradient	-0,059	0,064	0,358	-0.150	0.086	0.082
Phylogenetic diversity	Urban gradient	0,258	0,061	0,000	0.253	0.080	0.002
Population stability	Agricultural intensity gradient	0,047	0,083	0,575	-0.024	0.087	0.786
Population stability	Urban gradient	-0,366	0,089	0,000	-0.274	0.094	0.004
Population stability	Phylogenetic diversity	0,202	0,084	0,019	0.043	0.089	0.631
Population stability	Species diversity	0,133	0,087	0,131	0.147	0.092	0.113
Population asynchrony	Agricultural intensity gradient	0,176	0,086	0,042	0.097	0.083	0.246
Population asynchrony	Urban gradient	0,102	0,091	0,268	0.090	0.089	0.316
Population asynchrony	Phylogenetic diversity	0,257	0,087	0,004	0.242	0.085	0.005
Population asynchrony	Species diversity	0,283	0,090	0,002	0.411	0.088	0.000
Community stability	Population asynchrony	0,637			0.632		
Community stability	Population stability	0,710			0.734		

#5. In the abstract, the authors state that both population synchrony and population stability have effects on community stability, but through different drivers, i.e. are important, but that diversity loss is mainly affecting synchrony and habitat degradation mainly decreases population stability. Yet, in the main text the authors mainly discuss the effects of habitat degradation on stability, and very little about diversity and synchrony. This inconsistency needs to be resolved.

We thanks the reviewer for this remark. We expanded our discussion of the relationships linking diversity to community stability lines 130-150:

“This extends classical results found for experimental plant communities^{4,5} to animal taxa with very different natural history and thereby suggests that the positive biodiversity-stability relationship found for primary production applies to other functions and services provided by animal communities. Our results also bring new support to the phylogenetic insurance hypothesis²⁰, with a positive effect of phylogenetic diversity on population asynchrony found for bat and butterfly communities. This suggests that for these taxa, related species tend to share the traits involved in their response to environmental variations and perturbations and that we can use phylogeny as a proxy to assess the dynamical response of their populations to such environmental variations and perturbations.”

#6. The different environmental drivers are estimated at different spatial scales. Landscape variables are assessed within radii of 250, 500 and 1000 m for bird and butterflies and in squares

of 2-3 km for birds. In contrast, agricultural intensity variables are assessed with administrative
regions (of unclear size, but probably much larger than the areas for which the landscape
variables area assessed). This means that it is problematic to conclude that “while butterfly
community stability is mainly impacted by urbanization, bird and bat communities are mostly
destabilized by agricultural intensification...”. In reality, it might instead be that butterfly
communities are impacted by (any) drivers at smaller spatial scales while bird and bat
communities are impacted by (any) drivers at larger spatial scales. Also, the radii of 250-1000 m
seems quite small for butterflies an especially for bats, which are flying over rather large.

We think that there is a misunderstanding regarding the interpretation of our analysis of the
environmental drivers, this probably due to a lack of explanation. While we assessed the composition
of the landscape surrounding our study sites with buffers of different radii, we also characterised the
intensity of the agricultural practices using an index available at the regional level (of average size of
29000 km²). We calculated this last index as the amount of spending in agricultural input (fertilizers,
pesticides, biocides) divided by the agricultural area of the administrative region. Although the spatial
resolution of this index is not very high (but data with higher resolution are not available), it gives
information about the average intensity of agricultural practices within the region which is still
meaningful for France as some regions have clearly more intensive agricultural practices than others.

This variable of intensity of agricultural practices, as well as the landscape composition (proportion of
the different land-uses within the buffer) were included in the principal component analysis (PCA) for
all buffer radii. Therefore, this agricultural intensity variable allows differentiating sites with the same
landscape composition (proportion of the different land-uses within the buffer) but in different
administrative region. The site in the region using more agricultural input would then be position
further away along the PCA axis and would therefore be considered as having a surrounding landscape
with more intensive agriculture.

The two landscape gradients extracted from the principal component analysis (PCA) indicate that in
the set of sites we studied, we have sites with various level of urbanization (first axis of the PCA) and
this for all radii. Independently of this gradient, our monitored sites vary in the amount of intensive
crop land surrounding them (second axis of the PCA), and this for all radii too (see figure 2). The best
radius for each taxa is then selected based on AIC criteria. Therefore, in our analysis, there is no
possible confusion between the scale at which the landscape composition affects the community
dynamic and the nature of the landscape gradient that affects the community dynamic. In other words,
even if agricultural inputs are averaged at the French administrative region, it does not mean that this
variable reflect an effect on community variability at the regional scale, but it informs on the amount
of agricultural inputs that the cropland around the site and quantified within our buffers received.

To clarify these points we expended the explanation of the method (lines 425-466), modified the main
text lines 63 to 70 and integrated the supplementary figure 1 in the main text with expended legend.

Regarding the radii used for bats and butterflies, we choose them based on the literature. Flick et al.
(2012) and Krauss et al. (2003), showed that the spatial extent at which landscape has the strongest
effects on butterflies richness and abundance was between 250m. This relative small spatial scale is
correlated with the average daily movement distance of butterflies, between 200 and 600. Regarding
bats, we followed Azam et al., (2016) who studied the impact of landscape on bats with the same data
as us, and used buffers of 200m, 500m, 700m and 1000m radii.

Flick, T., Feagan, S. & Fahrig, L. Effects of landscape structure on butterfly species richness and
abundance in agricultural landscapes in eastern Ontario, Canada. *Agriculture, Ecosystems and*
*Environment* **156**, 123–133 (2012).

Krauss, J., Steffan-Dewenter, I., Tscharnke, T., 2003. How does landscape context contribute to
effects of habitat fragmentation on diversity and population density in butterflies? *J. Biogeogr.* 30,
889–900.

Azam, C., Le Viol, I., Julien, J.-F., Bas, Y. & Kerbiriou, C. Disentangling the relative effect of light pollution, impervious surfaces and intensive agriculture on bat activity with a national-scale monitoring program. *Landscape Ecology* **31**, 2471–2483 (2016).

Minor comments:

#7. I would like to see something about methods or the type of data used in the Abstract.

We added the following sentence:

“Here we analyse the relationships among landscape composition, biodiversity and community stability looking at time series of three types of communities, i.e. bats, birds and butterflies, monitored over the years by citizen science programs..”

#8. Lines 285-286: “...gardens that had been monitored in July...” Does this meant that only data from July was included in the analyses? But several of the butterfly species are active mainly during other parts of the year??

We restricted butterfly data to July to get the highest number of followed sites with the same months of observations. This criterion allowed us to combine a good amount of data without heterogeneity in the data due to variations in butterfly species abundance among months between species with different phenologies. Moreover, the 14 species used in this study do have flying adults during July.

#9. Line 377: Why did you use Corine Land cover data from 2006, and not more recent data (which I believe should be available).

See our response to comment # 3 reviewer 1, we have now updated our landscape analyses to include more recent data as well.

Reviewers' comments:

Reviewer #1 (Remarks to the Author):

In this revision, the authors addressed my comments and concerns. I appreciate this paper looking at the mechanisms by which landscape changes affect animal communities.

Reviewer #2 (Remarks to the Author):

The authors have presented a well written and analyzed study on what determines community instability. The authors well addressed my previous comments.

Reviewer #3 (Remarks to the Author):

The revision has significantly improved the quality of the manuscript, but in my view there are still some outstanding issues.

I maintain that I think that the time series is short for analysing community stability over time. Invertebrates such as butterflies are known to fluctuate at least an order of magnitude in population size between years, and this also affects community dynamics. The argument that other studies also have used short time series does not solve this issue.

My previous comment that only a part of the habitat was sampled (for butterflies) is also not fully resolved. The authors argue that population declines should be reflected in gardens even if butterflies only forage (and not reproduce) there. But the point is that the extent to which butterflies use gardens for foraging might differ between years, depending on e.g. weather. In a dry year, an irrigated garden might be used to a much higher extent than natural habitats, but this might not be true in another year.

After careful consideration, my colleagues and I agreed that the two outstanding concerns raised by Reviewer 3 on the study duration and the representativeness of the habitat sampling for butterflies have been partly addressed already. In particular, we are inclined to agree that the effects of “boom-bust” dynamics of single species should be dampened in your community-level analysis, as you argue in the response letter. However, it is important that this point is more clearly expressed in the main text, along with other caveats on potential issues regarding 1) the duration (for all taxa), and 2) potential biases of the sampling scheme (especially for the butterflies).

To clarify the limitations of our study, we added a paragraph lines 174 to 190 in the "limit and conclusion" section part, discussing the impact of the time series duration, in particular in relation to the estimation of population and community variability, and the potential biases related to the butterfly sampling scheme:

“Limits and conclusion

Here we measured community stability at a relatively short-time scale (up to 6, 17 and 11 years for respectively bat, bird and butterfly communities), reflecting the time scale used in most studies on the relationship between diversity and community stability⁶. However, population and community temporal variability are known to increase with the considered time scale⁴⁰⁻⁴² and as such, our estimates of temporal variability might underestimate the full variability of the studied communities. While this should not affect the effects of landscape composition we found, and indeed our results are robust when compared with analyses on two subsets of our datasets with different time series durations (see Methods and Supplementary Fig. 7-9), longer time series would be required to estimate the full variability of the studied communities. Another limitation of our study is that we assessed habitat degradation at the landscape scale and did not account for local conditions, such as management practices, that could also affect community variability. For example, butterfly data were collected in private gardens with different management strategies that are known to affect the attractiveness for butterflies⁴³. Accounting for such management practices as well as other local scale characteristics such as habitat heterogeneity that is also known to affect population stability¹⁶ would improve our understanding of the determinant of community stability.”

We also ask that you take special care to ensure that statements throughout the manuscript do not inflate the strength of the results, and that the terminology with respect to population and community stability is not misleading, e.g. the statement in L157-159 of the manuscript with tracked changes suggests that population stability of single species was measured separately from the community data rather than partitioned from the latter using the approach from reference 22. (Please note that the Methods section does not contribute to the word limit, and therefore could and should be expanded as needed to ensure that readers can follow it.).

We thank you for raising this issue. We carefully checked the terminology with respect to population and community stability, adding “weighted mean” to “population stability” to make clear that we talk about the component of the community stability used in the statistical analyses. This change was performed lines 12, 78, 114, 123, 125.

As explained above, we added a full paragraph about the limitation of our study and further made some change to not overstate our results, for example lines 100 and 145-146.

Finally, we ask that you clarify why the data and codes are only available upon request and were not shared for the review process. At a minimum, we ask that you provide the raw data underlying the key figures, please see the "Data Deposition" and "Source Data" paragraphs below.

We modified the data and codes availability in the manuscript, and we uploaded our data and codes in a Zenodo repository [<https://doi.org/10.5281/zenodo.3678366>].

In addition to these changes, during the revision process, we noticed several typos that we corrected as well as some unclear parts that we tried to clarify. This includes typos in the values presented in fig.3; some

clarifications of the axis labels in the supplementary figures and in the legends of the figures; a clearer explanation of the effects of landscape composition on population stability.

Reviewer #3

I maintain that I think that the time series is short for analysing community stability over time. Invertebrates such as butterflies are known to fluctuate at least an order of magnitude in population size between years, and this also affects community dynamics. The argument that other studies also have used short time series does not solve this issue.

See our answer to editor's comment above and the insertion made in the limit section of the manuscript, line 175-184:

“Here we measured community stability at a relatively short-time scale (up to 6, 17 and 11 years for respectively bat, bird and butterfly communities), reflecting the time scale used in most studies on the relationship between diversity and community stability⁶. However, population and community temporal variability are known to increase with the considered time scale⁴⁰⁻⁴² and as such, our estimates of temporal variability might underestimate the full variability of the studied communities. While this should not affect the effects of landscape composition we found, and indeed our results are robust when compared with analyses on two subsets of our datasets with different time series durations (see Methods and Supplementary Fig. 7-9), longer time series would be required to estimate the full variability of the studied communities.”

My previous comment that only a part of the habitat was sampled (for butterflies) is also not fully resolved. The authors argue that population declines should be reflected in gardens even if butterflies only forage (and not reproduce) there. But the point is that the extent to which butterflies use gardens for foraging might differ between years, depending on e.g. weather. In a dry year, an irrigated garden might be used to a much higher extent than natural habitats, but this might not be true in another year.

See our answer to editor's comment above and the insertion made in the limit section of the manuscript, line 184-190:

Another limitation of our study is that we assessed habitat degradation at the landscape scale and did not account for local conditions, such as management practices, that could also affect community variability. For example, butterfly data were collected in private gardens with different management strategies that are known to affect the attractiveness for butterflies⁴³. Accounting for such management practices as well as other local scale characteristics such as habitat heterogeneity that is also known to affect population stability¹⁶ would improve our understanding of the determinant of community stability.